# Vision-Enhanced Time Series Forecasting by Decomposed Feature Extraction and Composed Reconstruction

## Abstract

Time series forecasting plays a crucial role in various domains, such as power and weather forecasting. In recent years, different types of models have achieved promising results in long-term time series forecasting. However, these models often produce predictions that lack consistency with the style of the input, resulting in reduced reliability and trust in the forecasts. To address this issue, we propose the Vision-Enhanced Time Series Forecasting by Decomposed Feature Extraction and Composed Reconstruction (VisiTER), which leverages the rich semantic information provided by the image modality to enhance the realism of the predictions. It consists of two main components: the Decomposed Time Series to Image Generation and the Composed Image to Time Series Generation. In the first component, the Decomposed Time Series Feature Extraction Model extracts periodic and trend information, which is then transformed into images using our proposed time series to vision transformation architecture. After converting the input time series into images, the resulting images are used as style features and concatenated with the previously extracted features. In the second component, we use our proposed TimeIR along with the previously obtained feature set to perform image reconstruction for the prediction part. Due to the rich information provided, the reconstructed images exhibit better consistency with the input images, which are then transformed back into time series. Extensive experiments on seven real-world datasets demonstrate that VisiTER achieves state-of-the-art prediction performance on both traditional metrics and new metrics.

## 1 Introduction

Time series forecasting has always been a widely studied research area (Lim & Zohren, 2021; Torres et al., 2021), with extensive applications in fields such as economics (Granger & Newbold, 2014), energy (Qian et al., 2019; Martín et al., 2010), and weather forecasting (Wu et al., 2023b). The goal of time series forecasting is to predict future values based on past observed data. Most existing time series prediction methods focus on extracting periodicity, anomalies, random fluctuations, and other features from the time series data. However, relying solely on the information provided by the time series modality is often insufficient, leading to limited prediction accuracy and an inability to efficiently capture complex relationships within the data (Ismail Fawaz et al., 2019).

One promising strategy for addressing this issue is to convert time series into images for prediction (Li et al., 2024; Yang et al., 2024; Hatami et al., 2018). This is because the image modality proposes a new perspective for data modeling compared with time series, providing more rich and diverse information. Nevertheless, there are several key challenges in using images for time series prediction tasks. The first challenge is the difficulty in image transformation and model training. Existing transformation methods directly map time series into scatter plots, which can result in information loss in the images, ultimately leading to poor performance of the subsequent image models. The complexity of image models also results in prolonged training and inference times, consuming substantial GPU memory and increasing the difficulty of training. This is particularly severe for time series with many variables, which correspond to the number of channels in the image version. The second is ineffective image utilization in time series: The advantages and methodologies for effectively utilizing the image modality have not been adequately explored. Some existing prediction

methods that use images rely on image generation for forecasting. However, the generated images often lack fidelity in the context of time series, as they miss crucial temporal feature information.

To address these challenges, we propose **Visi**on-Enhanced **T**ime Series Forecasting by Decomposed Feature **E**xtraction and Composed **R**econstruction (**VisiTER**). It introduces a novel framework of time series forecasting by transformation and generation between time series and vision domains, which consists of two main components: (1) the Decomposed Time Series to Vision Generation, which extracts decomposed temporal features as images for further prediction; (2) the Composed Vision to Time Series Generation, which utilizes transformed images to generate composed prediction results by image reconstruction.

In the first component, we propose two main modules: the Decomposed Time Series Feature Extraction (DTFE) and the Time Series to Vision Transformation (T2V). DTFE decomposes the original time series and leverages the respective strengths of different types of transformer-based models to extract two essential temporal features for forecasting: the periodic and trend features. These decomposed features help the model utilize richer temporal information and improve prediction accuracy, which benefits the following image generation and reconstruction processes. Then, T2V adopts a novel approach to convert time series into images, mapping data points with diminishing pixel values along the y-axis. Utilizing T2V, we transform features from DTFE into image modality, improving the time series data distribution in corresponding images for better reconstruction.

In the second component, we introduce TimeIR, a novel transformer-based image reconstruction model specifically designed for time series data, to generate the forecasting results by image generation. By utilizing the periodic features, trend features extracted from DTFE, and style features of the time series as priors, TimeIR can enrich the reconstruction process with valuable temporal information, fully utilizing the potential of vision models. Notably, incorporating style information allows the reconstructed time series to retain the same style as the input time series. To address the challenges of difficult training and inference, we redesign the model architecture and adopt a unique training strategy. Specifically, our model can perform segmented prediction for long sequences, which helps to reduce the computational load. During training, we also employ a channel sampling strategy to further decrease the computational requirements.

In conclusion, our work makes the following key contributions:

- We propose the VisiTER framework, which enhances time series forecasting using image reconstruction models. By first extracting temporal features as priors, we provide the image model with rich information, thereby fully leveraging the advantages of the image modality.

- We propose DTFE, designed with various transformer-based models to extract periodic and trend features for more realistic representations. We also introduce T2V, effectively transforming time series into images, and TimeIR, which leverages priors for reconstructing time series images more efficiently.

- VisiTER achieves state-of-the-art performance on traditional MSE and MAE metrics. Moreover, we introduce the SSIM metric to time series forecasting tasks, enabling a more comprehensive evaluation of the structural integrity of predictions. On this metric, our model also outperforms other state-of-the-art models.

## 2 REALTED WORK

### 2.1 TIME SERIES FORECASTING

Time series forecasting methods can be categorized mainly into those based on Recurrent Neural Networks (RNNs)(Tokgöz & Ünal, 2018; Lai et al., 2018; Salinas et al., 2020), Convolutional Neural Networks (CNNs)(Wang et al., 2023; Hewage et al., 2020; Livieris et al., 2020), Transformers, and Multi-Layer Perceptrons (MLPs). In recent years, the Transformer model has emerged as a strong contender in time series forecasting (Liu et al., 2023; Vaswani, 2017; Zhou et al., 2021; Chen et al., 2024). Its self-attention mechanisms effectively capture both short-term and long-term dependencies, positioning it as a leading choice for many tasks. Linear models based on MLPs have also shown notable predictive results (Oreshkin et al., 2019; Challu et al., 2023; Wang et al., 2024), particularly for simpler datasets, serving as a useful baseline for comparison with more complex approaches.

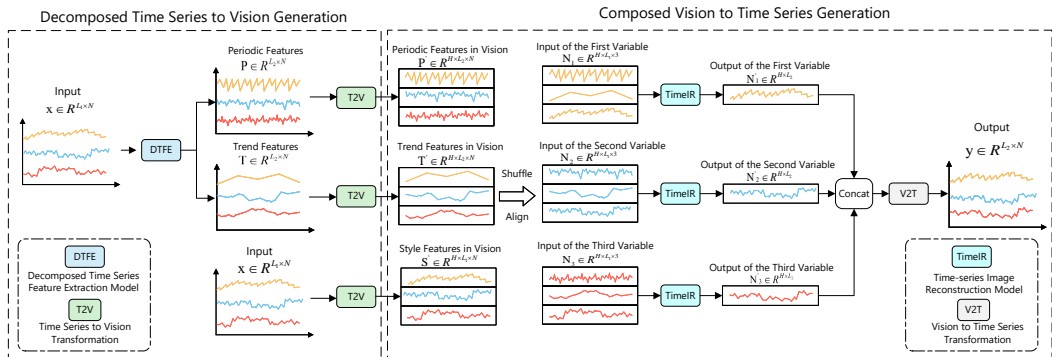

Figure 1: The overall architecture of VisiTER. Different colored sequences represent different variables in the time series, while the axes indicate the time series modality and the boxes represent the image modality. In the first part, we predict the periodic and trend features of the time series using the DTFE model, which are then converted into images. In the second part, we utilize TimeIR to reconstruct the images by integrating the periodic features, trend features, and style features, ultimately transforming the results back into time series data.

## 2.2 IMAGE RECONSTRUCTION

Image reconstruction (Park et al., 2003; Demoment, 1989) is a significant area in computer vision and image processing, aimed at recovering high-quality images from partial or degraded inputs. This task is essential in applications like medical imaging (Zhang & Dong, 2020), remote sensing (He et al., 2019), and video super-resolution (Kappeler et al., 2016; Liu et al., 2022a). Various methods have been proposed to tackle the image reconstruction problem, broadly categorized as follows. Optimization-based methods (Fessler, 2020): These establish mathematical models for reconstruction and use optimization algorithms, such as regularized optimization and dictionary learning. Deep learning-based methods (Liu et al., 2020; Liang et al., 2021): Utilizing deep neural networks for end-to-end reconstruction (Ledig et al., 2017; Sajjadi et al., 2017; Goodfellow et al., 2020), these methods, like SwinIR, effectively learn image priors from large datasets for high-quality results. GAN-based methods: Introducing the GAN framework, models like SRGAN and EnhanceNet generate more realistic and natural reconstructions.

## 2.3 IMAGE TECHNIQUES IN TIME SERIES

In existing time series-related tasks, images are primarily used for time series classification (Li et al., 2024; Dosovitskiy et al., 2021). The process typically involves converting time series data into images, followed by the application of traditional image models, such as Vision Transformers, to classify the images, thereby achieving classification of the time series. Additionally, there are extremely few models that utilize image models for time series forecasting (Yang et al., 2024), such as ViTime. These models generally decompose the time series into trend and periodic components and then generate subsequent images based on the provided input. However, this type of model does not fully leverage the rich information that the image modality offers, and they require long input lengths to provide information.

## 3 VISION-ENHANCED TIME SERIES FORECASTING FRAMEWORK

### 3.1 OVERALL ARCHITECTURE

We propose the overall framework as shown in the Figure 1. First, we are given the time series $\mathbf{X} \in R^{L_1 \times N}$, where $L_1$ and $N$ denote the look back length and the number of variates, which is input to the DTFE in Part I to obtain the trend feature $\mathbf{T} \in R^{L_2 \times N}$ and periodic feature $\mathbf{P} \in R^{L_2 \times N}$ of the time series, where $L_2$ denotes the prediction length. Next, $\mathbf{T}$, $\mathbf{P}$ and $\mathbf{x}$ are input into the T2V framework to be converted into image formats, and then they are shuffled based on different variables in Part II. For the same variable i, we concatenate the $\mathbf{P}_i$, $\mathbf{T}_i$, and $\mathbf{X}_i$. If $L_1$ is not equal to $L_2$, we also need to align them before concatenation. The subsequent TimeIR model operates on a univariate

basis, meaning we perform reconstruction for each variable individually. For a given time series, we segment it into N images for reconstruction, with each image corresponding to one of the N variables. Ultimately, we combine the reconstruction results back into a single image, which is then transformed back into the time series format.

## 3.2 DECOMPOSED TIME SERIES TO IMAGE GENERATION

In this section, we first predict the periodic information and trend features from the time series. We observe that the trend features of a variable are more susceptible to the influence of other variables, while its periodic information is less affected. Therefore, we propose the Decomposed Time Series Feature Extraction Model (DTFE), which employs a decomposed architecture using different Transformers to predict the periodic and trend features, resulting in more accurate outcomes.

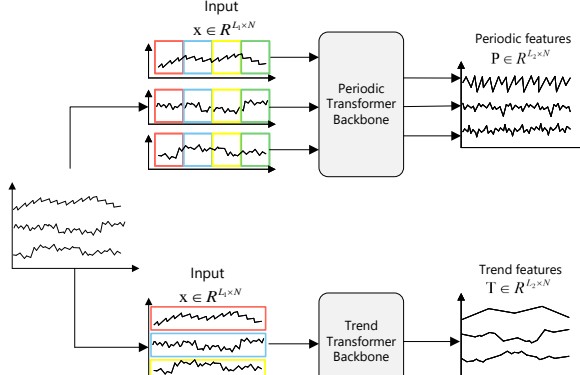

Next, we convert the predicted periodic information, trend information, and the input time series into images. The original direct mapping method transforms time series into scatter plots, which, while precise, has limited applicability for image reconstruction models due to the reduced number of activated pixels. This results in a sparse representation that does not fully leverage the potential of the image modality, lead-

Figure 2: Overall architecture of DTFE. Different variables are treated as tokens within the Transformer framework to predict trends, while multiple time steps of a single variable are considered as a token for predicting periodicity. Each row of the time series represents a distinct variable, while each differently colored box signifies a different token within the Transformer backbone. The specific architecture of the Transformer framework is illustrated in Figure 11 of the Appendix B.

ing to insufficient information for the reconstruction models to utilize effectively. To address this, we propose a new transformation method, T2V. We diffuse the scatter points, making the image continuous and more suitable for image models.

### 3.2.1 DECOMPOSED TIME SERIES FEATURE PREDICTION

In DTFE, for predicting the trend component, it is necessary to consider the inter-channel interactions more extensively, as the trend of one channel can be influenced by the trends of other channels. In contrast, for the periodic component, the influence of the periodicities in other channels on a given channel is relatively small, and hence, a more channel-independent approach can be adopted. Therefore, for the trend prediction model, we adopt an inverted transformer structure to explicitly capture the cross-channel dependencies. On the other hand, for the periodic component prediction, we use a patch-based transformer architecture, where we treat a few adjacent time steps as a patch and predict the periodicity based on this sequence of patches. By leveraging the strengths of both single-channel and multi-channel modeling approaches, and by tailoring the model structures to the specific characteristics of trend and periodicity, the proposed DTFE succeeds to achieve robust and accurate time series features prediction. The basic framework of DTFE is shown in the Figure 2.

When training DTFE, it is essential to compute the loss functions for the periodic Transformer backbone and the trend Transformer backbone separately for predicting the periodic and trend components, and then sum them together. For each periodic Transformer, denoted as P-Trans($\cdot$), we use $L_{\text{period}}$ to represent its MSE loss function, and for the trend Transformer, denoted as T-Trans($\cdot$), we use $L_{\text{trend}}$ to represent its MSE loss function. Additionally, we employ the Discrete Fourier Transform (DFT) to decompose the ground truth of forecasting result $\mathbf{y}$ into its period and trend components. Thus, we can derive the training losses for period and trend extraction:

$$L_{\text{period}} = \sum_{\mathbf{X}_i \in B} \left\lVert \text{P-Trans}(\mathbf{X}_i) - \text{DFT}(\mathbf{y}_i)_{\text{period}} \right\rVert_2^2 \tag{1}$$

Figure 3: The top row of the image illustrates the overall framework of T2V, which begins with normalization, followed by direct mapping to images, and then expands on both sides of the y-axis. The second row presents the overall framework of V2T, where the maximum value of each column is selected, mapped to a time series, and finally restored.

$$L_{\text{trend}} = \sum_{\mathbf{X}_i \in B} \left\| \text{T-Trans}(\mathbf{X}_i) - \text{DFT}(\mathbf{y}_i)_{\text{trend}} \right\|_2^2, \tag{2}$$

where $B$ represents the batch used for training, $\mathbf{y}_i$ denotes the ground truth of forecasting result of the input time series $\mathbf{X}_i$. Thus, we can achieve the overall loss $L_{\text{DTFE}} = L_{\text{period}} + L_{\text{trend}}$.

### 3.2.2 TIME SERIES TO IMAGE TRANSFORMATION

Given a time series $\mathbf{x} \in R^{L \times N}$, we need to transform it into an image $\mathbf{F} \in R^{H \times L \times N}$, where L corresponds to the length of the time series to be reconstructed, H is a hyperparameter that represents the height of the image after the transformation into a visual format and N denotes the number of variables involved. First, we normalize the time series $\mathbf{x}$ to have a variance of 1 and a mean of 0, to facilitate the subsequent operations: $\mathbf{x}' = \frac{\mathbf{x} - \mu(\mathbf{x})}{\sigma(\mathbf{x})}$, where $\mu(\mathbf{x})$ and $\sigma(\mathbf{x})$ represent the mean and standard deviation of $\mathbf{x}$ respectively. For each channel, we calculate the maximum value and divide the values in that channel by the maximum value. This maps the entire time series to the range of [-1, 1]: $\mathbf{x}'' = \frac{\mathbf{x}'}{max(\mathbf{x}', axis=0)}$. To control the range of time series within [0, H], we can multiply $\mathbf{x}''$ by $\frac{H}{2}$ and then add $\frac{H}{2}$: $\tilde{\mathbf{x}} = \mathbf{x}'' \times \frac{H}{2} + \frac{H}{2}$, which ensures that each value in the time series corresponds to the vertical coordinate values after conversion into an image format. Based on the normalized values, we can determine which pixels in the image F need to be activated:

$$\mathbf{F}_{i,j,c} = \begin{cases} 1, & \text{if } \tilde{\mathbf{x}}_{j,c} = i \\ 0, & \text{otherwise} \end{cases} \tag{3}$$

In this context, $\mathbf{F}_{i,j,c}$ denotes the pixel in $\mathbf{F}$ at the $i$-th row, $j$-th column, and $c$-th channel, while $\tilde{\mathbf{x}}_{j,c}$ represents the value of the $c$-th variable at the $j$-th time point in the time series $\tilde{\mathbf{x}}$. To help the reconstruction model perform better, we modify the scatter plot representation by extending the values on the Y-axis. Specifically, we enhance the neighboring pixels on either side of each activated pixel in a given column, with the value decreasing as the distance from the activated pixel increases. The extension range is [0,$\lambda$], where $\lambda$ is a hyperparameter to control extension ranges. The specific formula is as follows, where $k$ represents the current extension distance:

$$\mathbf{F}_{i,j,c} = \begin{cases} 1 - \frac{k}{\lambda}, & \text{if } \tilde{\mathbf{x}}_{j,c} = i \pm k \quad and \quad k \leq \lambda \\ 0, & \text{otherwise} \end{cases} \tag{4}$$

We transform the periodic features $\mathbf{P}$, trend features $\mathbf{T}$ and the input $\mathbf{x}$ into images using T2V respectively, resulting in $\mathbf{P}_{image}$, $\mathbf{T}_{image}$ and $\mathbf{X}_{image}$.

### 3.3 COMPOSED IMAGE TO TIME SERIES GENERATION

Based on the decomposed component images after prediction, we construct an image reconstruction model aimed at generating a complete image of the predicted results. To maintain consistency in style between the reconstructed predicted image and the original sequence, we introduce the image of the original sequence as additional style information. Consequently, we propose TimeIR, a temporal image reconstruction model, which first concatenates these three parts of input for the image reconstruction, then performs the reconstruction, and finally converts the results back into the time series format.

Figure 4: Alignment method of the model under different prediction lengths. When $L_1$ is greater than $L_2$, the length of $x$ is truncated. When $L_1$ equals $L_2$, no special processing is applied. When $L_1$ is less than $L_2$, a sliding window approach is used to segment and truncate $P$ and $T$ for prediction.

### 3.3.1 Composed Image Reconstruction

TimeIR is a univariate reconstruction model, so the following discussion pertains to the reconstruction of a single variable within a time series. The inputs we have received are $\mathbf{X}_{image} \in R^{H \times L_1 \times N}$, $\mathbf{P}_{image} \in R^{H \times L_2 \times N}$, and $\mathbf{T}_{image} \in R^{H \times L_2 \times N}$, which represent the original time series $\mathbf{x}$ as images, as well as the images depicting the periodicity and trends elucidated by the DTFE. The length of $\mathbf{X}_{image}$ is $L_1$, while the lengths of $\mathbf{P}_{image}$ and $\mathbf{T}_{image}$ are $L_2$. Next, we concatenate these three components to obtain the input $\mathbf{I} \in R^{H \times L \times 3}$, where the three components represent the three channels of $\mathbf{I}$. To perform the concatenation operation, it is necessary to align the lengths of the sequences. We handle different scenarios accordingly: when $L_1$ equals $L_2$, we proceed directly. When $L_1$ is greater than $L_2$, we truncate $\mathbf{X}_{image}$ to the length of $L_2$ since $\mathbf{X}_{image}$ serves merely as a style feature, and its length does not impact the information. When $L_1$ is less than $L_2$, we maintain a sliding window on $\mathbf{P}_{image}$ and $\mathbf{T}_{image}$ for learning. During each reconstruction, we select the periodic features and trend features within the sliding window, while keeping the style feature fixed at $\mathbf{X}_{image}$, as the style of the time series does not change with the sliding window selection. The reconstruction method is illustrated in Figure 4, and additional alignment details can be found in Appendix B.2. Thus, we can obtain style, trend, and periodicity priors, which we then concatenate to form the input to the TimeIR model.

The TimeIR model is comprised of three main components. First, a shallow feature extraction module utilizing a CNN network block is employed. This is followed by a deep feature extraction module, which consists of multiple Time Series Swin Transformer Blocks (TSTB). Each TSTB is composed of several Swin Transformer Blocks. The key characteristic of the Swin Transformer used in TimeIR is its utilization of an overlapping patch embedding scheme. This approach allows the model to better focus on the fine-grained details of the input data, which in this case is the time series. Finally, the deep feature representations are passed through a convolutional layer that reduces the channel dimension to 1, producing the final time series reconstruction. The overall architecture of TimeIR illustrated in Figure 10.

Since images and time series are different modalities, we need to convert the ground truth into images for training. We use MSE as the loss function between the predictions from TimeIR and the ground truth converted into images, as shown in the equation below.

$$L_{\text{TimeIR}} = \sum_{\mathbf{X}_i \in B} \left\| \text{TimeIR}(\mathbf{I}_i) - \text{T2V}(\mathbf{y}_i) \right\|_2^2 \tag{5}$$

In this equation, $L_{\text{TimeIR}}$ represents the loss function of training TimeIR, B represents the batch used for training, $\mathbf{y}_i$ denotes the ground truth of forecasting result of $\mathbf{X}_i$, and $\mathbf{I}_i$ refers to the combined information inputted into TimeIR for the time series $\mathbf{X}_i$.

### 3.3.2 Image to Time Series Transformation

The process of V2T is the inverse of the T2V procedure. Specifically, V2T converts the reconstruction results from TimeIR back into a time series, serving as the final prediction output. The first step is to identify the maximum value in each column of the image. The pixel corresponding to this maximum value is then set to 1, while all other pixels in that column are assigned a value of 0.

$$\mathbf{F}_{i,j,c} = \begin{cases} 1, & \text{if } \mathbf{F}_{i,j,c} = max(\mathbf{F}_{i,j,c}, axis = 0) \\ 0, & \text{otherwise} \end{cases} \tag{6}$$

Then, after rescaling the values to the range [-1, 1]. and normaling the values, we can get the predicted time series $\hat{\mathbf{y}}$. The complete T2V and V2T processes are illustrated in Figure 3.

## 4 EXPERIMENTS

**Datasets**   The datasets comprises 7 collections: ETT dataset (including 4 subsets:ETTh1, ETTh2, ETTm1, and ETTm2), Weather, Exchange, and Electricity datasets (Wu et al., 2021; Li et al., 2021). A detailed description of the dataset can be found in the Appendix A.1.

**Baseline**   We will compare VisiTER with 11 latest baselines, including PatchTST (Nie et al., 2023), iTransformer (Liu et al., 2023), Crossformer (Zhang & Yan, 2023), Autoformer (Wu et al., 2021), SparseTSF (Lin et al., 2024), TimesNet (Wu et al., 2023a), DLinear (Zeng et al., 2023), FEDformer (Zhou et al., 2022) , Non-Stationary Transformers (Liu et al., 2022c), SCINet (Liu et al., 2022b), and TiDE (Das et al., 2023).

**Main results**   The comprehensive forecasting results are listed in Table 1, with the best results marked in red and the second-best blue, while the visual results are displayed in Figure 5. In the case where the prediction length is greater than 96, our VisiTER model uses the model trained on the prediction length of 96 for the dataset to perform zero-shot prediction. We can see that our model achieve SOTA results on multiple datasets, especially performing particularly well on datasets with fewer variables. Furthermore, we find that the image reconstruction module is able to maintain low traditional MSE and MAE evaluation metrics, while also preserving the reconstruction style.

Table 1: Multivariate forecasting results with prediction lengths $S \in \{96, 192, 336, 720\}$ for others and fixed lookback length $T = 96$. Results are averaged from all prediction lengths. TimeIR performs **ZERO-SHOT** inference when predicting lengths of $\{192, 336, 720\}$. *Avg* means further averaged by subsets. Full results are listed in Table 5.

| Models | VisiTER (Ours) | | iTransformer (2023) | | SparseTSF (2024) | | PatchTST (2023) | | Crossformer (2023) | | TiDE (2023) | | TimesNet (2023a) | | DLinear (2023) | | SCINet (2022b) | | FEDformer (2022) | | Stationary (2022c) | |
|---|---|---|---|---|---|---|---|---|---|---|---|---|---|---|---|---|---|---|---|---|---|---|
| Metric | MSE | MAE | MSE | MAE | MSE | MAE | MSE | MAE | MSE | MAE | MSE | MAE | MSE | MAE | MSE | MAE | MSE | MAE | MSE | MAE | MSE | MAE |
| ETTm1 | **0.386** | **0.371** | 0.407 | 0.410 | 0.416 | 0.408 | 0.387 | 0.400 | 0.513 | 0.496 | 0.419 | 0.419 | 0.400 | 0.406 | 0.403 | 0.407 | 0.485 | 0.481 | 0.448 | 0.452 | 0.481 | 0.456 |
| ETTm2 | **0.279** | **0.323** | 0.288 | 0.332 | 0.288 | 0.329 | 0.281 | 0.326 | 0.757 | 0.610 | 0.358 | 0.404 | 0.291 | 0.333 | 0.350 | 0.401 | 0.571 | 0.537 | 0.305 | 0.349 | 0.306 | 0.347 |
| ETTh1 | **0.431** | **0.428** | 0.454 | 0.447 | 0.440 | 0.429 | 0.469 | 0.454 | 0.529 | 0.522 | 0.541 | 0.507 | 0.458 | 0.450 | 0.456 | 0.452 | 0.747 | 0.647 | 0.440 | 0.460 | 0.570 | 0.537 |
| ETTh2 | **0.368** | **0.398** | 0.383 | 0.407 | 0.383 | 0.402 | 0.387 | 0.407 | 0.942 | 0.684 | 0.611 | 0.550 | 0.414 | 0.427 | 0.559 | 0.515 | 0.954 | 0.723 | 0.437 | 0.449 | 0.526 | 0.516 |
| ECL | 0.194 | 0.285 | 0.178 | **0.270** | 0.224 | 0.297 | 0.205 | 0.290 | 0.244 | 0.334 | 0.251 | 0.344 | 0.192 | 0.295 | 0.212 | 0.300 | 0.268 | 0.365 | 0.214 | 0.327 | 0.305 | 0.349 |
| Exchange | **0.334** | **0.389** | 0.360 | 0.403 | 0.361 | 0.408 | 0.367 | 0.404 | 0.940 | 0.707 | 0.370 | 0.413 | 0.416 | 0.443 | 0.354 | 0.414 | 0.750 | 0.626 | 0.519 | 0.429 | 0.193 | 0.296 |
| Weather | **0.254** | **0.277** | 0.258 | 0.278 | 0.276 | 0.294 | 0.259 | 0.281 | 0.259 | 0.315 | 0.271 | 0.320 | 0.259 | 0.287 | 0.265 | 0.317 | 0.292 | 0.363 | 0.309 | 0.360 | 0.288 | 0.314 |

### 4.1 TRAINING STRATEGY

To avoid the excessively high computational demands of traditional image reconstruction models, we have adopted several strategies during the training process of TimeIR.

Firstly, for a given dataset, we only train the TimeIR for the case of 96-length prediction and 96-length input. When the prediction length is greater than 96, we directly use this trained model for zero-shot prediction, and have achieved very good results. Secondly, since some of the datasets have a large number of variables, we employ a channel sampling operation during training. Specifically, for each training batch, we select a different set of variables to train on. Lastly, we employed a sliding window strategy to align the input, ensuring that when the prediction length exceeds the input length, the size of TimeIR's input remain fixed, so the GPU memory consumption is independent of the prediction length. This allows our model to handle even very long prediction lengths without running into issues of infeasibility. The training strategy for the entire VisiTER framework is described in detail in Appendix A.2.

### 4.2 SSIM

The SSIM is a metric used to measure the similarity between images (Wang et al., 2004). Unlike traditional pixel-level error metrics, such as MSE, SSIM places greater emphasis on the structural information and perceptual quality of the images. The fundamental idea behind SSIM is to evaluate

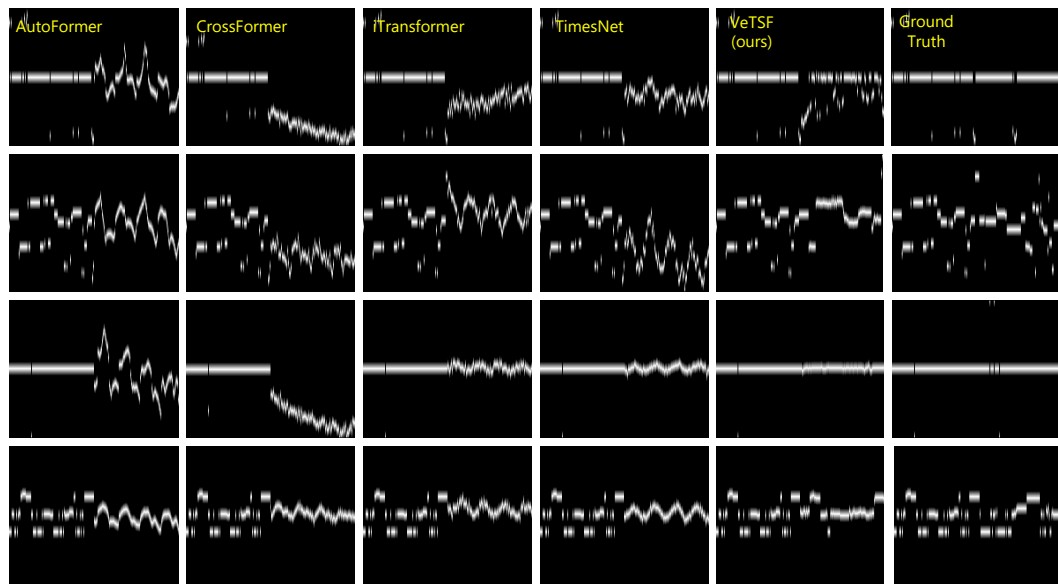

Figure 5: Comparison of visual results for time series reconstruction. The experiment focuses on predicting 96 steps with a lookback length of 96, using the ETTh2 dataset, where results from different variables are sampled. Each row represents a sample and each column represents a method. The first half of each image displays the provided time series, while the second half shows the predictions from various models. The last column represents the ground truth. Please zoom in for a closer view.

Table 2: The SSIM results of our model compared to other baselines with prediction lengths $S \in \{96, 192, 336, 720\}$ for others and fixed lookback length $T = 96$. Results are averaged from all prediction lengths, with the best results highlighted in bold. TimeIR performs **ZERO-SHOT** inference when predicting lengths of $\{192, 336, 720\}$. Higher SSIM values indicate better performance. The complete SSIM results are presented in Table 6.

| Model | ETTm1 | ETTm2 | ETTh1 | ETTh2 | ECL | Exchange | Weather |
|---|---|---|---|---|---|---|---|
| Autoformer (Wu et al., 2021) | 0.3762 | 0.4836 | 0.3867 | 0.4101 | 0.6051 | 0.6168 | 0.4377 |
| Crossformer (Zhang & Yan, 2023) | 0.3800 | 0.3822 | 0.4047 | 0.2620 | 0.6788 | 0.3417 | 0.4199 |
| iTransformer (Liu et al., 2023) | 0.4329 | 0.5025 | 0.4171 | 0.4163 | **0.7015** | 0.6443 | 0.5837 |
| TimesNet (Wu et al., 2023a) | 0.4292 | 0.5119 | 0.4072 | 0.4013 | 0.6486 | 0.6203 | 0.5720 |
| VisiTER **(ours)** | **0.4553** | **0.5384** | **0.4606** | **0.4395** | 0.6868 | **0.6597** | **0.5941** |

the quality of two images by comparing their similarity in terms of luminance, contrast, and structure. For time series images generated by our T2V method, brighter pixel values represent a higher likelihood, meaning that luminance, contrast, and structure all reflect the authenticity of the time series. SSIM values range from -1 to 1, where 1 indicates that the two images are identical, while 0 or negative values suggest a low level of similarity. A detailed introduction to SSIM and its corresponding formulas can be found in Appendix A.4.

We evaluated the performance of our model on various datasets using the SSIM metric. For comparison, we selected four state-of-the-art baselines: Autoformer, Crossformer, TimesNet, and iTransformer. Given that the $\lambda$ in T2V can impact the SSIM values, we have set $\lambda$=100 to facilitate the comparison. The results are presented in Table 2. It can be observed that our VisiTER model outperforms the other models by a significant margin in terms of SSIM across multiple datasets. This demonstrates that the time series predicted by our model exhibits stronger fidelity and structural similarity compared to the baselines.

### 4.3 ABLATIONS

#### 4.3.1 EFFECTIVENESS OF DTFE STRUCTURE.

To evaluate the effectiveness of our proposed DTFE architecture, we have conducted comparisons on the ETTh2 dataset against other Transformer-based models, including PatchTST and iTransformer. By directly summing the predicted trends and periodic components from DTFE, we obtain an intermediate result of the VisiTER. Additionally, we have included another variant of the DTFE, which was trained using a different approach and the training strategy are presented in Figure 9. In our current training method, we separately train a periodic feature extractor and a trend feature extractor, and the loss function is the sum of the MSE of the periodic loss and the trend loss. The alternative training method involves directly adding the predicted periodic and trend components, and then computing the MSE loss between the aggregated prediction and the ground truth.

The results are presented in the Table 3, which the strategy A denotes calculating the loss separately, while the strategy B refers to calculating the loss after summing the components. We can observe that the DTFE trained using our composite-architecture outperforms the single-architecture Transformer models. Furthermore, the DTFE trained using strategy A demonstrates superior performance compared to the traditional aggregation-based training strategy B. This indicates that our model not only possesses greater interpretability but also achieves better results.

Table 3: Results of the ablation study on the effectiveness of DTFE on ETTh2.

| Models | 96 | 192 | 336 | 720 | Avg |
|---|---|---|---|---|---|
| iTransformer | 0.297 | 0.380 | 0.428 | 0.427 | 0.383 |
| PatchTST | 0.302 | 0.388 | 0.426 | 0.431 | 0.387 |
| DTFE(Strategy B) | 0.292 | 0.371 | 0.409 | 0.425 | 0.374 |
| DTFE(Strategy A) | **0.286** | **0.366** | **0.404** | **0.420** | **0.370** |

#### 4.3.2 ABLATION OF $\lambda$

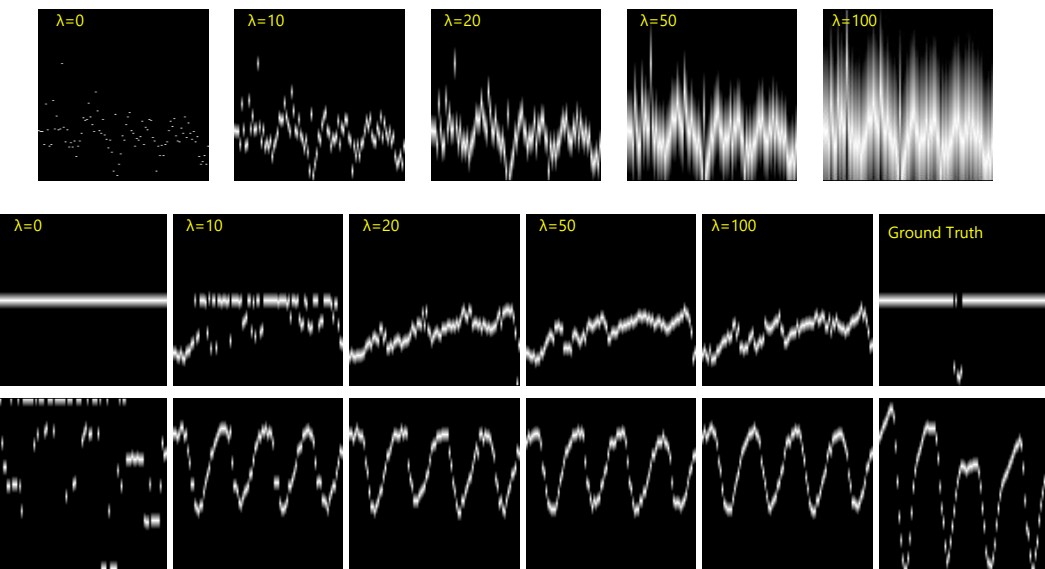

Figure 6: The first row is the visualization of the same time series under different $\lambda$. Others are the results obtained from training with different $\lambda$ values during the training process, with the last column representing the ground truth. For better visibility, the visualized results only capture the prediction portion in this figure.

In order to compare the influence of different degree of expansion in T2V ($\lambda$) on the performance of TimeIR, We conducted experiments with an input length of 96 and a prediction length of 96 on the ETTh2 dataset. In this study, we have fixed the DTFE and ensured that all the inputs are the same,

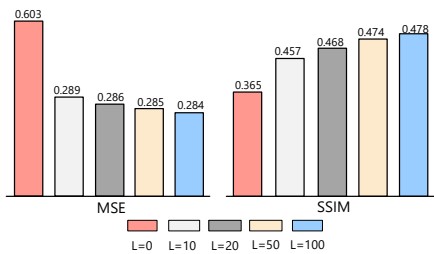 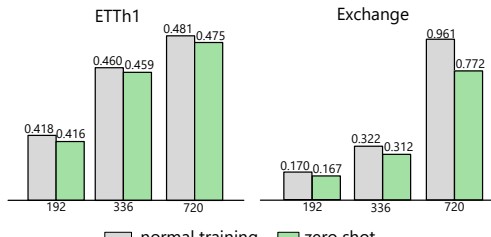

Figure 7: Results of the ablation study on the $\lambda$ of TimeIR on ETTh2.

Figure 8: Results of the ablation study on the zero-shot capability.

while also maintaining consistent training hyperparameters. Furthermore, the SSIM values reported here are measured with $\lambda=100$ for the sake of uniform comparison. The visualization results are presented in Figure 6, and the experimental results are shown in Figure 7.

It can be observed that as the value of $\lambda$ increases, both the MSE and SSIM metrics exhibit better performance. However, the visual inspection of a few sample cases reveals that when $\lambda$ is smaller, the reconstructed models tend to be more extreme, while larger values of $\lambda$ result in smoother reconstructions. In other words, smaller values of $\lambda$ generate results that are more stylistically similar to the real data, but this similarity is not necessarily reflected in the numerical metrics. Thus, $\lambda$ can be considered a hyperparameter that alters the model's ability to fuse styles. A smaller $\lambda$ leads to reconstructed time series that are more similar in style to the input, but at the cost of ignoring the overall coherence of the time series. Conversely, a larger $\lambda$ results in less influence from the style of the input sequence.

### 4.3.3 ZERO-SHOT ANALYSIS

In our experiments, to reduce the computational cost of training, we have only conducted training the TimeIR on the same dataset for sequence length 96 with 96-step prediction. For other cases, we have adopted a zero-shot approach. For prediction lengths greater than the sequence length, our training strategy involves randomly selecting the starting position of the sliding window for training. Additionally, we keep the DTFE model fixed while maintaining the same hyperparameters for the others. The datasets used in this experiment are Weather and ETTh1, and the results are presented in the Figure 8.

The results indicate that the zero-shot performance surpasses that of the standard training approach, and this difference becomes more pronounced as the prediction length increases. The underlying reason for this seemingly counterintuitive result is related to the design strategy of our model. Our model is designed to use the input sequence to provide the style features of the time series. However, when the prediction length is significantly longer than the sequence length, the style or characteristics may change. When the sliding window is close to the starting point, the style is more similar, but as the window moves further away, the style can become less similar. This can lead to training instability and poorer performance compared to the zero-shot approach.

## 5 CONCLUSION

We propose the Vision-Enhanced Time Series Forecasting by Decomposed Feature Extraction and Composed Reconstruction Framework (VisiTER) that leverages image reconstruction techniques for time series forecasting. The framework consists of two main components: the Decomposed Time Series to Image Generation and the Composed Image to Time Series Generation. It successfully integrates the image modality into time series forecasting. By supplementing the time series data with rich information from the image modality, the prediction results become more reliable and accurate. In future work, we hope to see a growing interest in exploring the use of image modalities within the field of time series forecasting. This integration could uncover new avenues for enhancing predictive models and improving performance.

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

## A  IMPLEMENTATION DETAILS

### A.1  DATASET DESCRIPTIONS

We conduct experiments on seven real-world datasets to evaluate the performance of the proposed VisiTER, including:

- ETTh1,ETTh2: This dataset (Kim et al., 2021) contains electricity transformer data recorded hourly from July 2016 to July 2018. It includes various operational factors that influence electricity consumption and transformer performance.

- ETTm1,ETTm2: This dataset (Kim et al., 2021) consists of electricity transformer measurements recorded every 15 minutes over the same time span. The higher frequency of data points allows for analysis of more granular trends and patterns in electricity usage.

- Exchange (Wu et al., 2021): This dataset compiles daily exchange rate panel data from eight countries, covering the period from 1990 to 2016. It includes various currency pairs, providing a rich resource for studying financial time series and the effects of economic events on exchange rates.

- Weather: This dataset (Wu et al., 2021) features 21 meteorological factors, such as temperature, humidity, and wind speed, collected every 10 minutes from the Weather Station of the Max Planck Biogeochemistry Institute in 2020. It serves as a vital resource for understanding climate patterns and their relationship with other time-dependent variables.

- ECL: This dataset records (Wu et al., 2021) hourly electricity consumption data from 321 clients. It provides insights into consumer behavior and demand patterns, making it useful for load forecasting and energy management studies.

We follow the same data processing and train-validation-test set split protocol used in TimesNet, ensuring a strict chronological order to prevent data leakage. For forecasting settings, we fix the lookback series length at 96 for all datasets, while the prediction length varies among {96, 192, 336, 720}. Detailed information about the datasets is provided in Table 4.

### A.2  TRAINING PROCESS

In the training of our entire framework, we begin by training the DTFE Model. Subsequently, we freeze its parameters and proceed to train the TimeIR model. Specifically, we focus on training the portion of the dataset that corresponds to a prediction length of 96. For any other prediction lengths, we directly employ the model for zero-shot inference.

### A.3  EXPERIMENT DETAILS

All experiments were conducted on an NVIDIA RTX 4090. We employed the ADAM optimizer and MSE as the loss function. The learning rate for all experiments was set at 0.0001. In Part 1, during the training of DTFE, a batch size of 32 was selected, while in Part 2, the batch size for training TimeIR was set to 5. Both Transformer blocks in DTFE consist of a single layer, whereas the TSTB in TimeIR has two layers.

Table 4: Dataset detailed descriptions. The dataset size is organized in (Train, Validation, Test).

| Dataset | Dim | Series Length | Dataset Size | Frequency | Information |
|---|---|---|---|---|---|
| ETTm1 | 7 | {96, 192, 336, 720} | (34465, 11521, 11521) | 15min | Temperature |
| ETTm2 | 7 | {96, 192, 336, 720} | (34465, 11521, 11521) | 15min | Temperature |
| ETTh1 | 7 | {96, 192, 336, 720} | (8545, 2881, 2881) | 15 min | Temperature |
| ETTh2 | 7 | {96, 192, 336, 720} | (8545, 2881, 2881) | 15 min | Temperature |
| Exchange | 8 | {96, 192, 336, 720} | (5120, 665, 1422) | Daily | Economy |
| Electricity | 321 | {96, 192, 336, 720} | (18317, 2633, 5261) | Hourly | Electricity |
| Weather | 21 | {96, 192, 336, 720} | (36792, 5271, 10540) | 10 min | Weather |

Figure 9: Different training strategies for DTFE. (a) Utilizing different types of Transformer models to separately predict periodicity and trends. (b) A conventional model ensemble method, where predictions are directly summed to compute the loss function.

### A.4 METRIC DETAILS

Regarding metrics, we utilize the mean square error (MSE) and mean absolute error (MAE) for forecasting. The calculations of these metrics are:

$$\text{MSE} = \sum_{i=1}^{F}(\mathbf{X}_i - \widehat{\mathbf{X}}_i)^2, \qquad\qquad \text{MAE} = \sum_{i=1}^{F}|\mathbf{X}_i - \widehat{\mathbf{X}}_i|,$$

where $\mathbf{X}, \widehat{\mathbf{X}} \in \mathbb{R}^{F \times C}$ are the ground truth and prediction results of the future with $F$ time pints and $C$ dimensions. $\mathbf{X}_i$ means the $i$-th future time point. At the same time, we convert the time series into images for Structural Similarity Index (SSIM) testing (Wang et al., 2004). The SSIM relies on three relatively autonomous components: luminance, contrast, and structures. It is widely recognized and better aligned with the requirements of perceptual assessment. SSIM estimates the luminance $\mu_{\mathbf{y}}$ of an image $\mathbf{y}$ as the mean of the intensity, while it estimates the contrast $\sigma_{\mathbf{y}}$ as the standard deviation of the intensity.

$$\mu_{\mathbf{y}} = \frac{1}{N_{\mathbf{y}}} \sum_{p \in \Omega_{\mathbf{y}}} \mathbf{y}_p, \tag{7}$$

$$\sigma_{\mathbf{y}} = \frac{1}{N_{\mathbf{y}} - 1} \sum_{p \in \Omega_{\mathbf{y}}} [\mathbf{y}_p - \mu_{\mathbf{y}}]^2 \tag{8}$$

To enable the comparison of these entities, a similarity comparison function $S$ is introduced:

$$S(x, y, c) = \frac{2 \cdot x \cdot y + c}{x^2 + y^2 + c}, \tag{9}$$

The variables $x$ and $y$ are the scalar variables being compared, $c = (k \cdot L)^2$, where $0 < k \ll 1$ is a constant used to avoid instability. Given a real image $\mathbf{y}$ and its approximation $\hat{\mathbf{y}}$, the comparison for brightness ($\mathcal{C}_l$) and contrast ($\mathcal{C}_c$) is as follows:

$$\mathcal{C}_l(\mathbf{y}, \hat{\mathbf{y}}) = S(\mu_{\mathbf{y}}, \mu_{\hat{\mathbf{y}}}, c_1) \text{ and } \mathcal{C}_c(\mathbf{y}, \hat{\mathbf{y}}) = S(\sigma_{\mathbf{y}}, \sigma_{\hat{\mathbf{y}}}, c_2) \tag{10}$$

where $c_1, c_2 > 0$. The empirical co-variance

$$\sigma_{\mathbf{y}, \hat{\mathbf{y}}} = \frac{1}{N_{\mathbf{y}} - 1} \sum_{p \in \Omega_{\mathbf{y}}} (\mathbf{y}_p - \mu_{\mathbf{y}}) \cdot (\hat{\mathbf{y}}_p - \mu_{\hat{\mathbf{y}}}), \tag{11}$$

determines the structure comparison ($\mathcal{C}_s$), expressed as the correlation coefficient between $\mathbf{y}$ and $\hat{\mathbf{y}}$:

$$\mathcal{C}_s(\mathbf{y}, \hat{\mathbf{y}}) = \frac{\sigma_{\mathbf{y}, \hat{\mathbf{y}}} + c_3}{\sigma_{\mathbf{y}} \cdot \sigma_{\hat{\mathbf{y}}} + c_3}, \tag{12}$$

where $c_3 > 0$. Finally, the SSIM is defined as:

$$\text{SSIM}(\mathbf{y}, \hat{\mathbf{y}}) = [\mathcal{C}_l(\mathbf{y}, \hat{\mathbf{y}})]^{\alpha} \cdot [\mathcal{C}_c(\mathbf{y}, \hat{\mathbf{y}})]^{\beta} \cdot [\mathcal{C}_s(\mathbf{y}, \hat{\mathbf{y}})]^{\gamma} \tag{13}$$

where $\alpha > 0, \beta > 0$ and $\gamma > 0$ are adjustable control parameters for weighting relative importance of all components.

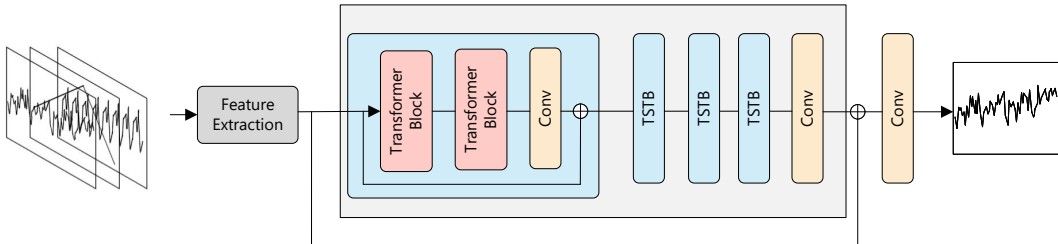

Figure 10: Overall architecture of TimeIR. Initially, shallow feature extraction is performed, followed by deep extraction using multiple TSTB layers, with the Transformer architecture detailed in Figure 11

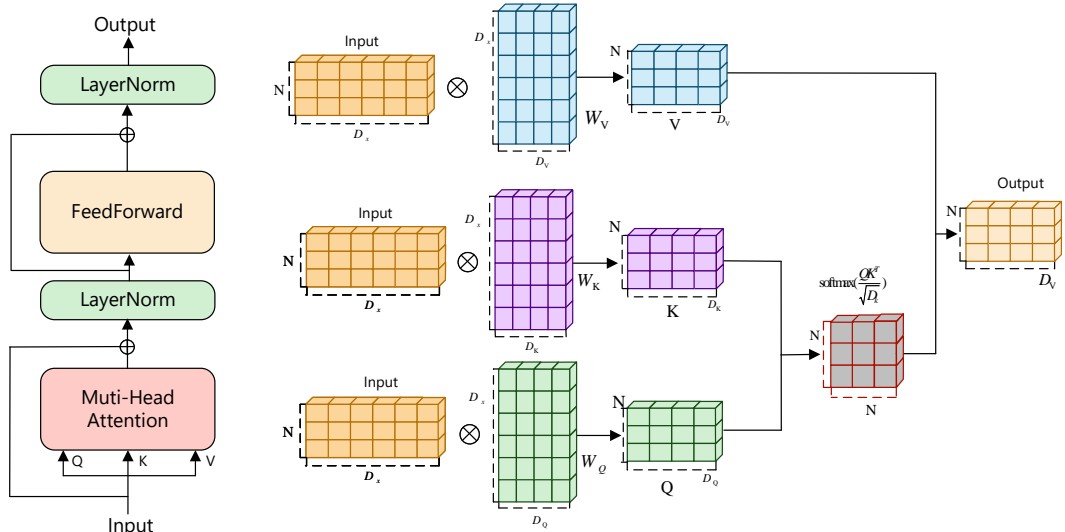

Figure 11: The overall architecture of the Transformer backbone. The left side of the figure illustrates the basic architecture of the Transformer model, while the right side presents a schematic representation of the self-attention mechanism.

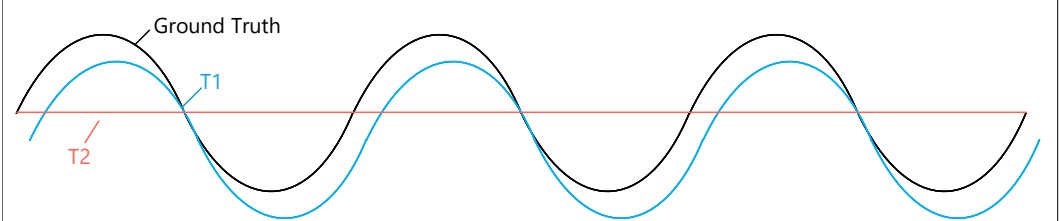

Figure 12: The black line represents the ground truth, the blue line denotes Time Series One, and the red line indicates Time Series Two.

## B  MODEL DETAILS

### B.1  IMAGE RECONSTRUCTION IN TIME-SERIES FORECASTING

The reason we introduced image reconstruction techniques in time series prediction tasks is that current time series prediction models often lack style continuity. Specifically, the geometric structure of the predicted time series does not align with that of the input time series. This issue arises from the use of the MSE loss function, which only reflects the numerical similarity between the predicted results and the true results, without capturing structural similarity. In other words, two samples may have the same MSE with respect to the ground truth, yet their shapes may differ.

In Figure 12, we present a detailed example: the ground truth is a sine function, while Time Series One is a sine function that has been shifted both horizontally and vertically, and Time Series Two is a straight line. By controlling the magnitude of the shift in Time Series One, we can make both time series have the same MSE as the ground truth. However, it is clear that Time Series Two lacks any meaningful information, such as periodicity and trends, even though its MSE is identical to that of Time Series One. In practical applications of time series, we would prefer to use Time Series Two, which possesses similar periodicity and trends and contains more information.

When ordinary time series prediction models use MSE as the loss function, they focus solely on numerical similarity while neglecting the geometrical structural similarity in a two-dimensional space. Therefore, we attempted to introduce image reconstruction models to capture this aspect of information. Experimental results have shown that our model achieves better reconstruction results, maintaining a low MSE while providing improved structural integrity.

### B.2  ALIGNMENT DETAILS

To ensure the alignment of $\mathbf{P}$, $\mathbf{T}$ and $\mathbf{X}$ for concatenation in Part II, we must match their lengths. When the input length is less than the prediction length, we truncate $\mathbf{X}$ to the input length. This approach is effective because $\mathbf{x}$ serves as a style feature, and altering its length does not significantly impact the stored information. When the input length equals the prediction length, concatenation can be done directly. When the prediction length exceeds the input length, we maintain a sliding window on $\mathbf{P}$ and $\mathbf{T}$ with a length equal to the input length. The window moves from the beginning to the end, capturing segments that are then concatenated with $\mathbf{X}$ to predict the corresponding output segments. For any excess length after division, an additional window is used to capture the final portion. The reason all windows are concatenated with $\mathbf{X}$ is that $\mathbf{X}$ acts as a style feature and does not provide any temporal information. Algorithm 1 outlines the complete operation of the model, including a detailed explanation of the input alignment logic.

## C  FULL FORECASTING RESULTS

Table 5 presents a comprehensive comparison of VisiTER with other baselines across seven datasets. It is evident that our model performs exceptionally well on the majority of the datasets.

---

**Algorithm 1** VisiTER - Overall Architecture.

---

**Require:** Input lookback time series $\mathbf{X} \in \mathbb{R}^{L_1 \times N}$; input Length $L_1$; predicted length $L_2$; variates
number $N$; $H$ is a hyperparameter in T2V; $X_i$ represents the image version of $X$

1: $\mathbf{P}, \mathbf{T} = \text{DTFE}(\mathbf{X})$.          $\triangleright \mathbf{P} \in \mathbb{R}^{L_2 \times N}, \mathbf{T} \in \mathbb{R}^{L_2 \times N}$

2: $\mathbf{P_i} = \text{T2V}(\mathbf{P})$.          $\triangleright \mathbf{P_i} \in \mathbb{R}^{H \times L_2 \times N}$

3: $\mathbf{T_i} = \text{T2V}(\mathbf{P})$.          $\triangleright \mathbf{T_i} \in \mathbb{R}^{H \times L_2 \times N}$

4: $\mathbf{X_i} = \text{T2V}(\mathbf{X})$.          $\triangleright \mathbf{X_i} \in \mathbb{R}^{H \times L_1 \times N}$

5: **if $\mathbf{L_1 \geq L_2}$:**

6:      **for $n$ in $\{0, \ldots, \mathbf{N} - 1\}$:**      $\triangleright$ Processing each variable individually

7:          $\mathbf{H_n} = \mathbf{Concat}(X_i[:, : L_2, n], P_i[:, :, n], T_i[:, :, n])$      $\triangleright \mathbf{H_n} \in \mathbb{R}^{H \times L_2 \times 3}$

8:          $\triangleright$ Image reconstruction using the TimeIR model

9:          $\mathbf{I_n} = \text{TimeIR}(\mathbf{H_n})$      $\triangleright \mathbf{I_n} \in \mathbb{R}^{H \times L_2}$

10:      $\mathbf{I} = \mathbf{Concat}(\mathbf{I_1}, \ldots, \mathbf{I_n})$      $\triangleright \mathbf{I} \in \mathbb{R}^{H \times L_2 \times N}$

11: **else $\mathbf{L_1 \geq L_2}$:**

12:      **for $l$ in $\{0, \ldots, \mathbf{L_2}//\mathbf{L_1} - 2\}$:**      $\triangleright$ Sliding window.

13:          **for $n$ in $\{0, \ldots, \mathbf{N} - 1\}$:**      $\triangleright$ Processing each variable individually

14:            $\triangleright \mathbf{H_n^l} \in \mathbb{R}^{H \times L_1 \times 3}$

15:            $\mathbf{H_n^l} = \mathbf{Concat}(\mathbf{X_i}, \mathbf{P_i}[:, l \times \mathbf{L_1} : (l+1) \times \mathbf{L_1}, \mathbf{n}], \mathbf{T_i}[:, l \times \mathbf{L_1} : (l+1) \times \mathbf{L_1}, \mathbf{n}])$

16:            $\mathbf{I_n} = \text{TimeIR}(\mathbf{H_n^l})$      $\triangleright \mathbf{I_n} \in \mathbb{R}^{H \times L_1}$

17:      $\mathbf{I^l} = \mathbf{Concat}(\mathbf{I_1}, \ldots, \mathbf{I_n})$      $\triangleright \mathbf{I^l} \in \mathbb{R}^{H \times L_1 \times N}$

18:      **for $n$ in $\{0, \ldots, \mathbf{N} - 1\}$:**      $\triangleright$ Processing each variable individually

19:          $\triangleright \mathbf{H_n^{l+1}} \in \mathbb{R}^{H \times L_1 \times 3}$

20:          $\mathbf{H_n^{l+1}} = \mathbf{Concat}(\mathbf{X_{image}}, \mathbf{P_{image}}[:, -\mathbf{L_1} :, \mathbf{n}], \mathbf{T_{image}}[:, -\mathbf{L_1} :, \mathbf{n}])$

21:          $\mathbf{I_n} = \text{TimeIR}(\mathbf{H_n^{l+1}})$      $\triangleright \mathbf{I_n} \in \mathbb{R}^{H \times L_1}$

22:      $\mathbf{I^{l+1}} = \mathbf{Concat}(\mathbf{I_1}, \ldots, \mathbf{I_n})$      $\triangleright \mathbf{I^{l+1}} \in \mathbb{R}^{H \times L_1 \times N}$

23:      $\mathbf{I} = \mathbf{Concat}(\mathbf{I^0}, \ldots, \mathbf{I^l}, \mathbf{I^{l+1}}[:, -(\mathbf{L_2} \bmod \mathbf{L_1}) :, :])$      $\triangleright \mathbf{I} \in \mathbb{R}^{H \times L_2 \times N}$

24: $\hat{\mathbf{Y}} = \text{V2T}(\mathbf{I})$      $\triangleright \hat{\mathbf{Y}} \in \mathbb{R}^{L_2 \times N}$

25: **Return $\hat{\mathbf{Y}}$**      $\triangleright$ Return the prediction result $\hat{\mathbf{Y}}$

---

Table 5: Full results of the long-term forecasting task. We compare extensive competitive models under different prediction lengths following the setting of TimesNet (2023a). The input sequence length is set to 96 for all baselines. *Avg* means the average results from all four prediction lengths.

| Models | | VisiTER (Ours) | | iTransformer (2023) | | SparseTSF (2024) | | PatchTST (2023) | | Crossformer (2023) | | TiDE (2023) | | TimesNet (2023a) | | DLinear (2023) | | SCINet (2022b) | | FEDformer (2022) | | Stationary (2022c) | |
|---|---|---|---|---|---|---|---|---|---|---|---|---|---|---|---|---|---|---|---|---|---|---|---|
| Metric | | MSE | MAE | MSE | MAE | MSE | MAE | MSE | MAE | MSE | MAE | MSE | MAE | MSE | MAE | MSE | MAE | MSE | MAE | MSE | MAE | MSE | MAE |
| ETTm1 | 96 | **0.303** | **0.326** | 0.334 | 0.368 | 0.357 | 0.375 | 0.329 | 0.367 | 0.404 | 0.426 | 0.364 | 0.387 | 0.338 | 0.375 | 0.345 | 0.372 | 0.418 | 0.438 | 0.379 | 0.419 | 0.386 | 0.398 |
| | 192 | 0.369 | **0.354** | 0.377 | 0.391 | 0.394 | 0.393 | **0.367** | **0.385** | 0.450 | 0.451 | 0.398 | 0.404 | 0.374 | 0.387 | 0.380 | 0.389 | 0.439 | 0.450 | 0.426 | 0.441 | 0.459 | 0.444 |
| | 336 | 0.400 | **0.380** | 0.426 | 0.420 | 0.426 | 0.414 | **0.399** | 0.410 | 0.532 | 0.515 | 0.428 | 0.425 | 0.410 | 0.411 | 0.413 | 0.413 | 0.490 | 0.485 | 0.445 | 0.459 | 0.495 | 0.464 |
| | 720 | 0.473 | **0.423** | 0.491 | 0.459 | 0.488 | 0.449 | **0.454** | 0.439 | 0.666 | 0.589 | 0.487 | 0.461 | 0.478 | 0.450 | 0.474 | 0.453 | 0.595 | 0.550 | 0.543 | 0.490 | 0.585 | 0.516 |
| | Avg | **0.386** | **0.371** | 0.407 | 0.410 | 0.416 | 0.408 | 0.387 | 0.400 | 0.513 | 0.496 | 0.419 | 0.419 | 0.400 | 0.406 | 0.403 | 0.407 | 0.485 | 0.481 | 0.448 | 0.452 | 0.481 | 0.456 |
| ETTm2 | 96 | **0.174** | **0.255** | 0.180 | 0.264 | 0.186 | 0.268 | 0.175 | 0.259 | 0.287 | 0.366 | 0.207 | 0.305 | 0.187 | 0.267 | 0.193 | 0.292 | 0.286 | 0.377 | 0.203 | 0.287 | 0.192 | 0.274 |
| | 192 | **0.240** | **0.299** | 0.250 | 0.309 | 0.248 | 0.306 | 0.241 | 0.302 | 0.414 | 0.492 | 0.290 | 0.364 | 0.249 | 0.309 | 0.284 | 0.362 | 0.399 | 0.445 | 0.269 | 0.328 | 0.280 | 0.339 |
| | 336 | **0.302** | **0.340** | 0.311 | 0.348 | 0.308 | 0.343 | 0.305 | 0.343 | 0.597 | 0.542 | 0.377 | 0.422 | 0.321 | 0.351 | 0.369 | 0.427 | 0.637 | 0.591 | 0.325 | 0.366 | 0.334 | 0.361 |
| | 720 | **0.400** | **0.398** | 0.412 | 0.407 | 0.408 | 0.398 | 0.402 | 0.400 | 1.730 | 1.042 | 0.558 | 0.524 | 0.408 | 0.403 | 0.554 | 0.522 | 0.960 | 0.735 | 0.421 | 0.415 | 0.417 | 0.413 |
| | Avg | **0.279** | **0.323** | 0.288 | 0.332 | 0.288 | 0.329 | 0.281 | 0.326 | 0.757 | 0.610 | 0.358 | 0.404 | 0.291 | 0.333 | 0.350 | 0.401 | 0.571 | 0.537 | 0.305 | 0.349 | 0.306 | 0.347 |
| ETTh1 | 96 | **0.374** | **0.383** | 0.386 | 0.405 | 0.386 | 0.393 | 0.414 | 0.419 | 0.423 | 0.448 | 0.479 | 0.464 | 0.384 | 0.402 | 0.386 | 0.400 | 0.654 | 0.599 | 0.376 | 0.419 | 0.513 | 0.491 |
| | 192 | **0.416** | **0.422** | 0.441 | 0.436 | 0.435 | 0.422 | 0.460 | 0.445 | 0.471 | 0.474 | 0.525 | 0.492 | 0.436 | 0.429 | 0.437 | 0.432 | 0.719 | 0.631 | 0.420 | 0.448 | 0.534 | 0.504 |
| | 336 | **0.459** | 0.444 | 0.487 | 0.458 | 0.476 | **0.440** | 0.501 | 0.466 | 0.570 | 0.546 | 0.565 | 0.515 | 0.491 | 0.469 | 0.481 | 0.459 | 0.778 | 0.659 | 0.459 | 0.465 | 0.588 | 0.535 |
| | 720 | 0.475 | 0.461 | 0.503 | 0.491 | **0.460** | **0.455** | 0.500 | 0.488 | 0.653 | 0.621 | 0.594 | 0.558 | 0.521 | 0.500 | 0.519 | 0.516 | 0.836 | 0.699 | 0.506 | 0.507 | 0.643 | 0.616 |
| | Avg | **0.431** | **0.428** | 0.454 | 0.447 | 0.440 | 0.429 | 0.469 | 0.454 | 0.529 | 0.522 | 0.541 | 0.507 | 0.458 | 0.450 | 0.456 | 0.452 | 0.747 | 0.647 | 0.440 | 0.460 | 0.570 | 0.537 |
| ETTh2 | 96 | **0.284** | **0.337** | 0.297 | 0.349 | 0.304 | 0.347 | 0.302 | 0.348 | 0.745 | 0.584 | 0.400 | 0.440 | 0.340 | 0.374 | 0.333 | 0.387 | 0.707 | 0.621 | 0.358 | 0.397 | 0.476 | 0.458 |
| | 192 | **0.364** | **0.393** | 0.380 | 0.400 | 0.385 | 0.396 | 0.388 | 0.400 | 0.877 | 0.656 | 0.528 | 0.509 | 0.402 | 0.414 | 0.477 | 0.476 | 0.860 | 0.689 | 0.429 | 0.439 | 0.512 | 0.493 |
| | 336 | **0.406** | **0.423** | 0.428 | 0.432 | 0.421 | 0.428 | 0.426 | 0.433 | 1.043 | 0.731 | 0.643 | 0.571 | 0.452 | 0.452 | 0.594 | 0.541 | 1.000 | 0.744 | 0.496 | 0.487 | 0.552 | 0.551 |
| | 720 | **0.417** | 0.440 | 0.427 | 0.445 | 0.420 | **0.437** | 0.431 | 0.446 | 1.104 | 0.763 | 0.874 | 0.679 | 0.462 | 0.468 | 0.831 | 0.657 | 1.249 | 0.838 | 0.463 | 0.474 | 0.562 | 0.560 |
| | Avg | **0.368** | **0.398** | 0.383 | 0.407 | 0.383 | 0.402 | 0.387 | 0.407 | 0.942 | 0.684 | 0.611 | 0.550 | 0.414 | 0.427 | 0.559 | 0.515 | 0.954 | 0.723 | 0.437 | 0.449 | 0.526 | 0.516 |
| ECL | 96 | 0.170 | 0.263 | **0.148** | **0.240** | 0.210 | 0.280 | 0.181 | 0.270 | 0.219 | 0.314 | 0.237 | 0.329 | 0.168 | 0.272 | 0.197 | 0.282 | 0.247 | 0.345 | 0.193 | 0.308 | 0.169 | 0.273 |
| | 192 | 0.178 | 0.271 | **0.162** | **0.253** | 0.206 | 0.282 | 0.188 | 0.274 | 0.231 | 0.322 | 0.236 | 0.330 | 0.184 | 0.289 | 0.196 | 0.285 | 0.257 | 0.355 | 0.201 | 0.315 | 0.182 | 0.286 |
| | 336 | 0.194 | 0.287 | **0.178** | **0.269** | 0.219 | 0.296 | 0.204 | 0.293 | 0.246 | 0.337 | 0.249 | 0.344 | 0.198 | 0.300 | 0.209 | 0.301 | 0.269 | 0.369 | 0.214 | 0.329 | 0.200 | 0.304 |
| | 720 | 0.233 | 0.319 | 0.225 | **0.317** | 0.260 | 0.328 | 0.246 | 0.324 | 0.280 | 0.363 | 0.284 | 0.373 | **0.220** | 0.320 | 0.245 | 0.333 | 0.299 | 0.390 | 0.246 | 0.355 | 0.222 | 0.321 |
| | Avg | 0.194 | 0.285 | **0.178** | **0.270** | 0.224 | 0.297 | 0.205 | 0.290 | 0.244 | 0.334 | 0.251 | 0.344 | 0.192 | 0.295 | 0.212 | 0.300 | 0.268 | 0.365 | 0.214 | 0.327 | 0.193 | 0.296 |
| Exchange | 96 | **0.083** | **0.200** | 0.086 | 0.206 | 0.095 | 0.218 | 0.088 | 0.205 | 0.256 | 0.367 | 0.094 | 0.218 | 0.107 | 0.234 | 0.088 | 0.218 | 0.267 | 0.396 | 0.148 | 0.278 | 0.111 | 0.237 |
| | 192 | **0.167** | **0.293** | 0.177 | 0.299 | 0.184 | 0.307 | 0.176 | 0.299 | 0.470 | 0.509 | 0.184 | 0.307 | 0.226 | 0.344 | 0.176 | 0.315 | 0.351 | 0.459 | 0.271 | 0.315 | 0.219 | 0.335 |
| | 336 | 0.312 | 0.404 | 0.331 | 0.417 | 0.324 | 0.414 | **0.301** | **0.397** | 1.268 | 0.883 | 0.349 | 0.431 | 0.367 | 0.448 | 0.313 | 0.427 | 1.324 | 0.853 | 0.460 | 0.427 | 0.421 | 0.476 |
| | 720 | **0.772** | **0.660** | 0.847 | 0.691 | 0.839 | 0.691 | 0.901 | 0.714 | 1.767 | 1.068 | 0.852 | 0.698 | 0.964 | 0.746 | 0.839 | 0.695 | 1.058 | 0.797 | 1.195 | 0.695 | 1.092 | 0.769 |
| | Avg | **0.334** | **0.389** | 0.360 | 0.403 | 0.361 | 0.408 | 0.367 | 0.404 | 0.940 | 0.707 | 0.370 | 0.413 | 0.416 | 0.443 | 0.354 | 0.414 | 0.750 | 0.626 | 0.519 | 0.429 | 0.461 | 0.454 |
| Weather | 96 | 0.172 | **0.214** | 0.174 | **0.214** | 0.197 | 0.237 | 0.177 | 0.218 | **0.158** | 0.230 | 0.202 | 0.261 | 0.172 | 0.220 | 0.196 | 0.255 | 0.221 | 0.306 | 0.217 | 0.296 | 0.173 | 0.223 |
| | 192 | 0.218 | 0.255 | 0.221 | **0.254** | 0.244 | 0.273 | 0.225 | 0.259 | **0.206** | 0.277 | 0.242 | 0.298 | 0.219 | 0.261 | 0.237 | 0.296 | 0.261 | 0.340 | 0.276 | 0.336 | 0.245 | 0.285 |
| | 336 | 0.274 | 0.296 | 0.278 | 0.296 | 0.293 | 0.308 | 0.278 | 0.297 | **0.272** | 0.335 | 0.287 | 0.335 | 0.280 | 0.306 | 0.283 | 0.335 | 0.309 | 0.378 | 0.339 | 0.380 | 0.321 | 0.338 |
| | 720 | 0.351 | **0.345** | 0.358 | 0.347 | 0.368 | 0.357 | 0.354 | 0.348 | 0.398 | 0.418 | 0.351 | 0.386 | 0.365 | 0.359 | **0.345** | 0.381 | 0.377 | 0.427 | 0.403 | 0.428 | 0.414 | 0.410 |
| | Avg | **0.254** | **0.277** | 0.258 | 0.278 | 0.276 | 0.294 | 0.259 | 0.281 | 0.259 | 0.315 | 0.271 | 0.320 | 0.259 | 0.287 | 0.265 | 0.317 | 0.292 | 0.363 | 0.309 | 0.360 | 0.288 | 0.314 |

Table 6: The complete SSIM results of our model compared to other baselines with prediction lengths $S \in \{96, 192, 336, 720\}$ for others and fixed lookback length $T = 96$. TimeIR performs **ZERO-SHOT** inference when predicting lengths of $\{192, 336, 720\}$. *Avg* means further averaged by subsets. Higher SSIM values indicate better performance.

| Models | | ETTm1 | ETTm2 | ETTh1 | ETTh2 | ECL | Exchange | Weather |
|---|---|---|---|---|---|---|---|---|
| Autoformer (Wu et al., 2021) | 96 | 0.3829 | 0.4924 | 0.4070 | 0.4186 | 0.6110 | 0.6379 | 0.4381 |
| | 192 | 0.3498 | 0.4857 | 0.3518 | 0.4091 | 0.6112 | 0.6308 | 0.4191 |
| | 336 | 0.3729 | 0.4655 | 0.3936 | 0.4106 | 0.6061 | 0.6141 | 0.4379 |
| | 720 | 0.3990 | 0.4909 | 0.3945 | 0.4021 | 0.5655 | 0.5843 | 0.4557 |
| | Avg | 0.3762 | 0.4836 | 0.3867 | 0.4101 | 0.6051 | 0.6168 | 0.4377 |
| Crossformer (Zhang & Yan, 2023) | 96 | 0.3979 | 0.4594 | 0.4529 | 0.3826 | 0.6988 | 0.5692 | 0.4714 |
| | 192 | 0.3907 | 0.4247 | 0.4160 | 0.2423 | 0.6931 | 0.5137 | 0.3913 |
| | 336 | 0.3718 | 0.3610 | 0.3863 | 0.2163 | 0.6718 | 0.1772 | 0.4409 |
| | 720 | 0.3596 | 0.2837 | 0.3639 | 0.2070 | 0.6517 | 0.1067 | 0.3759 |
| | Avg | 0.3800 | 0.3822 | 0.4047 | 0.2620 | 0.6788 | 0.3417 | 0.4199 |
| iTransformer (Liu et al., 2023) | 96 | 0.4294 | 0.5232 | 0.4485 | 0.4409 | **0.7167** | 0.6971 | 0.6079 |
| | 192 | 0.4302 | 0.4991 | 0.4186 | 0.4167 | **0.7079** | 0.6561 | 0.5862 |
| | 336 | 0.4333 | 0.5029 | 0.4058 | 0.4077 | **0.6984** | 0.6328 | 0.5745 |
| | 720 | 0.4386 | 0.4848 | 0.3954 | 0.3998 | **0.6828** | 0.5911 | 0.5661 |
| | Avg | 0.4329 | 0.5025 | 0.4171 | 0.4163 | **0.7015** | 0.6443 | 0.5837 |
| TimesNet (Wu et al., 2023a) | 96 | 0.4267 | 0.5361 | 0.4394 | 0.4352 | 0.6626 | 0.6640 | 0.5884 |
| | 192 | 0.4237 | 0.5115 | 0.4206 | 0.3936 | 0.6520 | 0.6329 | 0.5725 |
| | 336 | 0.4259 | 0.5065 | 0.3935 | 0.3953 | 0.6453 | 0.6158 | 0.5637 |
| | 720 | 0.4405 | 0.4935 | 0.3754 | 0.3811 | 0.6345 | 0.5686 | 0.5633 |
| | Avg | 0.4292 | 0.5119 | 0.4072 | 0.4013 | 0.6486 | 0.6203 | 0.5720 |
| VisiTER (**ours**) | 96 | **0.4569** | **0.5624** | **0.4796** | **0.4772** | 0.6976 | **0.7156** | **0.6125** |
| | 192 | **0.4561** | **0.5410** | **0.4605** | **0.4178** | 0.6952 | **0.6723** | **0.5942** |
| | 336 | **0.4513** | **0.5286** | **0.4538** | **0.4310** | 0.6861 | **0.6456** | **0.5867** |
| | 720 | **0.4568** | **0.5216** | **0.4485** | **0.4319** | 0.6682 | **0.6053** | **0.5829** |
| | Avg | **0.4553** | **0.5384** | **0.4606** | **0.4395** | 0.6868 | **0.6597** | **0.5941** |

## D  MORE VISUALIZATION RESULTS

Figure 13 presents additional visual results. It is clear that our model provides more accurate predictions, particularly when the time series approaches a straight line. In such cases, our model is able to reconstruct it as a straight line, whereas traditional time series prediction models struggle to capture the fluctuations.

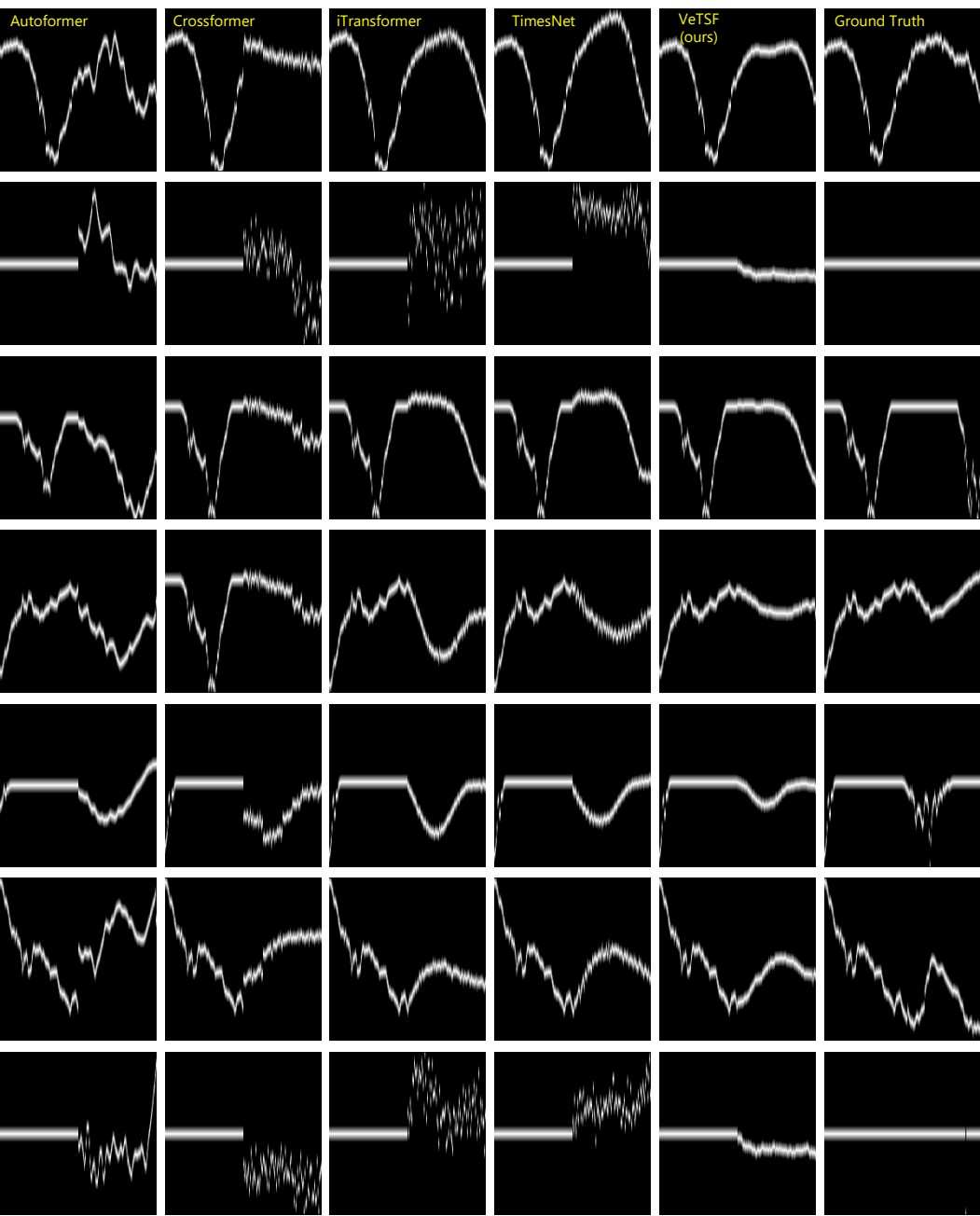

Figure 13: More comparison of visual results for time series reconstruction. The experiment focuses on predicting 96 steps with a lookback length of 96, using the ETTh2 and Weather dataset, where results from different variables are sampled. The first half of each image displays the provided time series, while the second half shows the predictions from various models. The last column represents the ground truth. Please zoom in for a closer view.

# E    EXPECTATIONS FOR FUTURE RESEARCH

This paper introduces the first method that utilizes image reconstruction for time series prediction, leveraging periodic and trend information to reconstruct time series. The core approach involves incorporating the style of the time series through image reconstruction. However, when predicting long time series, the style can change over time as the series lengthens, potentially introducing noise into the supplementary information. Future research could focus on better utilizing the style feature to address this issue.

Additionally, we directly employed traditional Transformer models for both feature prediction and time series image reconstruction. Subsequent work could modify the Transformer architecture to make it more suitable for reconstructing time series images. While our experiments concentrated on long time series prediction, this method is also applicable to other time series tasks, such as short time series prediction and imputation.

