# OpenReview forum: "Vision-Enhanced Time Series Forecasting by Decomposed Feature Extraction and Composed Reconstruction"
_ICLR.cc/2025/Conference — ICLR 2025 Conference Withdrawn Submission_

### Official Review · Reviewer_xofX · 2024-11-02

**Soundness:** 2
**Presentation:** 3
**Contribution:** 2
**Rating:** 3
**Confidence:** 4

**Summary:**

This paper presents VisiTER (Vision-Enhanced Time Series Forecasting), a new framework integrating image-based approaches to enhance time series forecasting. VisiTER consists of two main components: Decomposed Time Series to Image Generation and Composed Image to Time Series Generation. The approach transforms time series data into images to capture richer semantic information and generate consistent predictions with greater fidelity.

**Strengths:**

Good set of figures (fig 1, 2, 3) that roughly show the proposed idea

**Weaknesses:**

- Writing: The author should revise the comprehension and linking of ideas. For example:
    - line 47→55: Linking words choices (The first challenge … → Additionally, … → The second challenge …) can be improved.
    - line 47→49: Ambiguous connection: Why the complexity in image transformation can be explained with the latter idea of direct scatter plots causing information loss?
- Method:
    - from the text description (Sec. 3.2) and figures (1, 2), the idea is seem to linked with extraction of periodic and trend components of input data, while from the loss, the modules try to predict the these component for output instead. The authors should clarify this point and reflect it in the manuscript.
    - T2V module: why expanding in Y-dimension can help with reconstruction process?
- Experiment:
    - The overall pipeline involves DTFE, V2T which have their own transformer blocks inside. The author should have experiments calculating the computational overhead of proposed method, in comparison with its performance gain.

**Questions:**

Please refer to Weaknesses for related questions.

---

> ### Author Response · Authors · 2024-11-21
> **Response to Reviewer xofX**
>
> We would like to sincerely thank Reviewer xofX for providing a detailed review and insightful comments. Based on the suggestions, we have revised our paper accordingly.
>
> # W1:Use of linking words
>
> Thank you for your suggestions. We have made the corresponding modifications in the article to enhance its clarity and coherence.
>
> # W2:Ambiguous connection
>
> We apologize for any confusion caused by our choice of words. Here, "complexity" does not refer to computational complexity, but rather to the difficulty of constructing an appropriate method for transforming time series into images. Our intention is to convey that existing methods that directly map time series into scatter plots can lead to information loss, and designing a method that avoids such loss is challenging.
>
> # W3:Clarification of model objectives
>
> The reason we utilize the periodicity and trend of the predicted output is closely related to the characteristics of the image reconstruction model. Image reconstruction refers to the process of restoring a degraded image, where "degraded" indicates that the geometric structure of the image has been compromised. Given the limitations of image reconstruction models in capturing temporal dynamics, it is crucial to employ existing time series modeling techniques to predict the output's periodicity and trends. Consequently, the degraded image we maintain must reflect these predicted structural periodicity and trends. In other words, we need to use segments of the predicted results from the time series as the foundation for the image reconstruction process.
>
> # W4:Expanding in Y-dimension
>
> We need to extend the Y-axis because this adjustment provides a more accurate representation of the computed loss. Considering this scenario, when the predicted values are close to the actual values, in the time series modality, the corresponding loss should be small. However, in the image modality, these values do not correspond to the same pixel if the Y-axis is not expanded, resulting in a significantly larger loss in the image MSE, which does not align with our understanding and requirements regarding the errors in actual predictive results.
>
> By extending the Y-axis, we can ensure that as the predicted points get closer to the points in the ground truth (GT), the loss becomes smaller, which aligns with our expectations.
>
> # W5:The computational costs of proposed method
>
> In our model, the DTFE component includes a Transformer block, whereas the V2T component does not incorporate this block, as it serves as a conversion method that does not require training. We will first present the theoretical analysis of the computational complexity of the Transformer block in the DTFE component, where the sequence length is denoted as L, the patch size as P, the number of variables as N, and the number of Transformer layers as 1.
>
> When modeling with variables as tokens, the number of tokens is N and the dimensionality is L, resulting in a complexity of $O(N^2L)$. In contrast, when modeling using patches of time series, we consider patches as tokens, with the patch size defining the dimensionality, leading to a complexity of $O(\frac{L^2}{P})$. The computational complexity for the periodic prediction part is denoted as $O(\frac{L^2}{P})$, while that for the trend prediction part is denoted as $O(N^2L)$, resulting in an overall computational complexity for DTFE represented as $O(N^2L+\frac{L^2}{P})$.  We have selected commonly used models, iTransformer and patchTST, for comparison, with their respective computational complexities denoted as $O(2N^2L)$and  $O(2\frac{L^2}{P})$. The factor of 2 is included because both models have two layers.
>
> It is important to note that the specific values of these computational complexities are influenced by different datasets and prediction lengths. Therefore, we tested the computational complexity on a specific dataset, namely the ETTh2 dataset, with a prediction length of 96, using GMac as the unit of measurement for the computational complexity.
>
> | models       | MSE       | computational cost(GMac) |
> | ------------ | --------- | ------------------------ |
> | DTFE         | **0.286** | 0.08                     |
> | iTransformer | 0.297     | **0.02**                 |
> | PatchTST     | 0.302     | 0.14                     |
>
> Our model achieves the best prediction results, with computational costs positioned between the two provided baselines.

---

> ### Author Response · Authors · 2024-11-25
> **Response to Reviewer xofX**
>
> Dear Reviewer xofX,
>
> We would like to sincerely thank you for your time and efforts in reviewing our paper.
>
> We have made an extensive effort to try to address your concerns. In our response:
>
> - We revise the manuscript to reduce the use of inappropriate terminology and ambiguous expressions.
> - We provide a detailed explanation of why we use the output features as the prediction target.
> - We supplement our analysis with a comparison of the computational costs associated with our model and the baseline models.
>
> We hope our response can effectively address your concerns, If you have any further concerns or questions, please do not hesitate to let us know, and we will respond timely.
>
> All the best,
>
> Authors

---

> > ### Comment · Reviewer_xofX · 2024-11-30
> > **Reply to Authors**
> >
> > Dear the Authors,
> >
> > Thanks for your response addressing my comments.
> >
> > Your answer have cleared some of my initial concerns. However, the motivation of you using image reconstruction techniques to predict periodicity and trend of the future times series remain unclear for me. While these techniques are "good" or "suitable" to **reconstruct** corrupted images, I do not see why they are suitable for **prediction** task, especially for time series - a different modality. And with this choice, you now have to address **the limitations of image reconstruction models in capturing temporal dynamics** - cited the reply of the Authors.
> >
> > Given this, I believe this work can be further improved, and I would keep my score as it is.
> >
> > Thanks,
> >
> > Reviewer xofX

---

### Official Review · Reviewer_MNU9 · 2024-11-02

**Soundness:** 2
**Presentation:** 3
**Contribution:** 2
**Rating:** 5
**Confidence:** 3

**Summary:**

This paper introduces VisiTER, a novel method for time series forecasting that leverages the rich semantic information contained in images by converting time series data into visual representations. The proposed method consists of two primary components: the decomposition of time series data into images and the prediction of time series through three image inputs. In the first component, a feature extraction model decomposes the time series to isolate periodic and trend information, which is then transformed into images. In the second component, the authors employ the TimeIR module to perform time series forecasting via image reconstruction. Experimental results on seven real-world datasets demonstrate that VisiTER achieves state-of-the-art performance in both traditional and novel evaluation metrics.

**Strengths:**

1.Compared to traditional methods that directly map time series data to scatter plots and then reconstruct them, this paper proposes the T2V method. By diffusing and continuousizing the scatter points, the generated images are more suitable as inputs for image models. Additionally, the introduction of multiple Swin Transformer Blocks in the TimeIR model allows the model to better focus on the detailed information of the time series. This approach leverages the rich information in the image modality, enabling the model to capture complex relationships more effectively.

2.The paper introduces the DTFE module, which effectively extracts periodic and trend features, thereby enhancing the model's performance. This module can efficiently handle the periodic and trend characteristics of time series data and convert them into image form for reconstruction, improving the accuracy and stability of the reconstruction results.

3.The introduction of the TimeIR model enables more efficient utilization of periodic, trend, and style features for image reconstruction, which is crucial for time series prediction. This model enhances the overall predictive capabilities of the system.

4.Experimental results show that the VisiTER model achieves state-of-the-art performance across multiple datasets. Moreover, the authors use the SSIM metric to evaluate the similarity between the generated time series images and the actual time series. The results indicate that the VisiTER model also performs exceptionally well in terms of SSIM, demonstrating its robustness and effectiveness.

**Weaknesses:**

1.Lack of Motivation for Image Transformation. The authors do not provide sufficient justification for the advantages of converting time series data into images for the forecasting task. There is a lack of ablation studies to explore the motivation behind this transformation, which weakens the argument for its necessity.

2.The concept of "style features" in the context of time series tasks is somewhat abstract and not clearly defined. This lack of clarity makes it difficult for readers to understand the relevance and reasonableness of these features, potentially undermining the credibility of the approach.

3.The introduction of the SSIM metric seems to lack a strong rationale. While SSIM is useful for evaluating image quality, its relevance and significance in the context of time series prediction are not well justified. This raises concerns about the appropriateness of using such an image-based metric for this specific task.

4.The authors could enhance the robustness of their method by treating the image reconstruction component as a plug-in module and applying it to other existing baseline models. This would help to determine whether the image reconstruction approach can improve the performance of different models, providing stronger evidence for its effectiveness.

**Questions:**

Please refer to Weaknesses.

---

> ### Author Response · Authors · 2024-11-21
> **Response to Reviewer MNU9**
>
> We would like to sincerely thank Reviewer MNU9 for providing a detailed review and insightful comments. Based on the suggestions, we have revised our paper accordingly.
>
> # W1:The motivation for Image Transformation
>
> We employ image reconstruction techniques for processing time series data because current time series models fail to capture the inherent geometric structural information. These models typically use MSE or MAE between corresponding time points for training, which only reflects numerical similarity. For instance, two prediction results may yield the same MSE, yet their styles can differ significantly. Consider an example where the ground truth (GT) is a sine function  $y=sin(x)$ , Time Series One is a shifted sine function $y=sin(x+a)+b$, and Time Series Two is a straight line $y=0$. By adjusting the magnitude of the shift in Time Series One, Time Series One  and Time Series Two may have similar MSEs compared with GT. While the MSEs are similar, Time Series Two lacks geometric details and does not convey useful information such as periodicity and trends. Thus, low MSE alone is insufficient; we need to incorporate geometric structural similarity into our predictions.
>
> Since one-dimensional time serie sequences cannot capture two-dimensional geometric features well, we introduced image reconstruction components to our approach. This allows us to capture the combined structural similarity of time series in a two-dimensional context, where one dimension represents the time dimension and the other dimension represents the value dimension. Our experimental results demonstrate that our model can effciently capture these combined structures, as shown in the visualizations. More detailed information regarding the incorporation of images has been revised in the appendix, with further specifics available in Appendix B.1.
>
> We conduct ablation experiments on TimeIR using the ETTh1 dataset, and the specific results are as follows:
>
> | pred length | DTFE MSE | DTFE+TimeIR MSE | DTFE   MAE | DTFE+TimeIR   MAE | DTFE    SSIM | DTFE+TimeIR   SSIM |
> | ----------- | -------- | --------------- | ---------- | ----------------- | ------------ | ------------------ |
> | 96          | 0.377    | **0.374**       | 0.386      | **0.383**         | 0.4654       | **0.4796**         |
> | 192         | 0.420    | **0.416**       | 0.425      | **0.422**         | 0.4483       | **0.4605**         |
> | 336         | 0.461    | **0.459**       | 0.447      | **0.444**         | 0.4432       | **0.4538**         |
> | 720         | 0.478    | **0.475**       | 0.465      | **0.461**         | 0.4348       | **0.4485**         |
> | Avg         | 0.434    | **0.431**       | 0.431      | **0.428**         | 0.4479       | **0.4606**         |
>
> After adding TimeIR, it can be observed that our model achieve better prediction performance on the MSE metric., while the SSIM (which ranges from 0 to 1, with higher values indicating better performance) increased by 0.0127. This indicates that the TimeIR module not only enhances the geometric structural integrity of the time series but also reduces the traditional MSE loss.
>
> # W2:The concept of  style feature
>
> The style of the time series mentioned in the article can be defined as the geometric structural information of the time series.  Specific manifestations of style can be observed in our visualization results. In the example shown in the third row of Figure 5, the original input time series(the left half of each sequence) exhibits a linear style, while the ground truth for prediction (the right half of the sequence) also appears as a straight line. In contrast, the predictions made by other time series forecasting methods show fluctuating patterns, highlighting a clear stylistic difference from the original time series. This noticeable difference clearly indicates that the original time series and predicted time series do not constitute a unified time series, thus suggesting a discontinuity in their styles. However, our model successfully reconstructs the time series into a form that closely resembles a straight line, thereby preserving the style of the preceding series.

---

> ### Author Response · Authors · 2024-11-21
> **Response to Reviewer MNU9**
>
> # W3:The introduction of the SSIM metric
>
> We introduce the SSIM because existing evaluation metrics for time series lack a focus on structural assessment. Continuing with the previously mentioned example in response to W1, both Time Series One and Time Series Two have the same MSE value compared to the ground truth (GT), yet their geometric structures are entirely different. To evaluate their geometric structural similarity with the GT, we converted this example into images and conducted SSIM testing. Specifically, the SSIM value between Time Series One and the GT is 0.5866, while the SSIM value between Time Series Two and the GT is only 0.2722.
>
> This clearly indicates that Time Series One, which exhibits high geometric structural similarity with the GT, achieves a higher SSIM, whereas Time Series Two has a significantly lower SSIM. This demonstrates that SSIM effectively captures the geometric structural similarity of time series. Therefore, we introduce SSIM as an evaluation metric to supplement the limitations of MSE and MAE in assessing structural integrity.
>
> # W4:Treating the image reconstruction component as a plug-in module
>
> Our current image reconstruction model is built upon the joint reconstruction of the periodicity, trend, and features of the time series. In contrast, existing time series models predict only the time series itself, rather than their underlying cycles and trends. To transform our method into a plugin, we need to make modifications. Specifically, in TimeIR, we will continue to accept style inputs but replace the input features of periodicity and trends with the predictions from traditional time series models. We conduct experiments on iTransformer, selecting the ETTh2 dataset for our experiments. The specific experimental results are as follows:
>
> | pred length | iTransformer MSE | iTransformer+TimeIR MSE | iTransformer   MAE | iTransformer+TimeIR   MAE | iTransformer    SSIM | iTransformer+TimeIR   SSIM |
> | ----------- | ---------------- | ----------------------- | ------------------ | ------------------------- | -------------------- | -------------------------- |
> | 96          | 0.297            | **0.293**               | 0.349              | **0.344**                 | 0.4409               | **0.4569**                 |
> | 192         | 0.380            | **0.379**               | 0.400              | **0.399**                 | 0.4167               | **0.4247**                 |
> | 336         | 0.428            | **0.428**               | 0.432              | **0.431**                 | 0.4077               | **0.4134**                 |
> | 720         | 0.427            | **0.428**               | 0.445              | **0.445**                 | 0.3998               | **0.4053**                 |
> | Avg         | 0.383            | **0.382**               | 0.407              | **0.405**                 | 0.4163               | **0.4251**                 |
>
> It can be observed that after incorporating our TimeIR module, our model achieved a reduction of 0.01 in MSE, a reduction of 0.02 in MAE, and an increase of 0.0088 in SSIM. These results demonstrate that the proposed image reconstruction component provides improved predictions in accuracy and geometric structure of the time series.

---

> ### Author Response · Authors · 2024-11-25
> **Response to Reviewer MNU9**
>
> Dear Reviewer MNU9,
>
> We would like to sincerely thank you for your time and efforts in reviewing our paper.
>
> We have made an extensive effort to try to address your concerns. In our response:
>
> - We elucidate the motivation for using images in time series forecasting.
> - We provide a detailed explanation of the specific definition of style.
> - We present examples to analyze the rationale behind our introduction of SSIM.
> - We supplement our findings with corresponding experiments that incorporate our model as a plugin.
>
> We hope our response can address your concerns. If you have any further concerns or questions, please do not hesitate to let us know, and we will respond timely.
>
> All the best,
>
> Authors

---

### Official Review · Reviewer_pp94 · 2024-11-05

**Soundness:** 3
**Presentation:** 3
**Contribution:** 3
**Rating:** 5
**Confidence:** 4

**Summary:**

This work addresses time series forecasting from the view of image modal. The authors propose a novel model called VisiTER, which leverages image modality to enhance the realism and consistency of time series predictions. The experimental results show that VisiTER present promising performance.

**Strengths:**

1.	The paper introduces a novel approach to time series forecasting by framing the problem within the image modality, offering a fresh perspective.
2.	The model structure is clear and appears logically sound.
3.	The presentation is well-organized and easy to follow.

**Weaknesses:**

1.	The motivation behind the paper is that existing models often produce predictions that lack consistency with the "style" of the input. However, the concept of “style consistency” is not clearly defined, weakening the argument for this motivation.
2.	The rationale for using Vision Transformers (ViT) for time series feature extraction is not well-explained. Specifically, it is unclear what key differences between image and time series modalities justify the use of ViT. For instance, while the order/position of image patches is irrelevant, the temporal order of time series data is crucial. This distinction needs further elaboration.
3.	The paper omits some commonly-used datasets, such as Traffic and Solar-Energy. Including results from these datasets would strengthen the evaluation of the proposed model.
4.	The paper does not provide the source code, which is critical for reproducibility. Given that the reported results claim state-of-the-art performance, evidence supporting this claim is necessary.

**Questions:**

Please answer the concerns of W1 and W2, and supplement the materials of W3 and W4.

---

> ### Author Response · Authors · 2024-11-21
> **Response to Reviewer pp94**
>
> We would like to sincerely thank Reviewer pp94 for providing a detailed review and insightful comments. Based on the suggestions, we have revised our paper accordingly.
>
> # W1:The concept of  style consisency
>
> The continuity of style can be understood as a type of discrepancy measurement between the original time series and the predicted time series. Specifically, the style of the time series mentioned in the article can be defined as **the geometric structural information of the time series**. This can be illustrated by examining the visualization results in Figure 5. In the third row of examples, the original input time series(the left half of each sequence) exhibits a linear style, while the ground truth for prediction (the right half of the sequence) also appears as a straight line. In contrast, the predictions made by other time series forecasting methods show fluctuating patterns, highlighting a clear stylistic difference from the original time series. This noticeable difference clearly indicates that the original time series and predicted time series do not constitute a unified time series, thus suggesting a discontinuity in their styles. However, our model successfully reconstructs the time series into a form that closely resembles a straight line, thereby preserving the style of the preceding series.
>
> # W2(1):The motivation for Image Transformation in Time-Series Forecasting
>
> We employ image reconstruction techniques for processing time series data because current time series models fail to capture the inherent geometric structural information. These models typically use MSE or MAE between corresponding time points for training, which only reflects numerical similarity. For instance, two prediction results may yield the same MSE, yet their styles can differ significantly. Consider an example where the ground truth (GT) is a sine function  $y=sin(x)$ , Time Series One is a shifted sine function $y=sin(x+a)+b$, and Time Series Two is a straight line $y=0$. By adjusting the magnitude of the shift in Time Series One, Time Series One  and Time Series Two may have similar MSEs compared with GT. While the MSEs are similar, Time Series Two lacks geometric details and does not convey useful information such as periodicity and trends. Thus, low MSE alone is insufficient; we need to incorporate geometric structural similarity into our predictions.
>
> Since one-dimensional time series sequences cannot capture two-dimensional geometric features well, we introduced image reconstruction components to our approach. This allows us to capture the combined structural similarity of time series in a two-dimensional context, where one dimension represents the time dimension and the other dimension represents the value dimension. Our experimental results demonstrate that our model can effciently capture these combined structures, as shown in the visualizations.
>
> # W2(2):The distinction between Time series and image
>
> In the image modality, the order of each patch is significant; altering this order results in changes to the image, similar to the characteristics of time series data. Additionally, our TimeIR model incorporates positional embeddings to ensure that the model comprehends the sequence of the patches. The primary distinction between these two modalities is that the image modality, while preserving the information of the time series, introduces a y-axis, thereby providing additional geometric structural information and enhancing the capability to capture style. More detailed information regarding the incorporation of images has been revised in the appendix, with further specifics available in Appendix B.1.

---

> ### Author Response · Authors · 2024-11-21
> **Response to Reviewer pp94**
>
> # W3:Missing dataset
>
> Thank you for your reminder. We have supplemented evaluation of our model on these two datasets on MSE metric, and the results are as follows:
>
> | dataset          | ours      | Crossformer | iTransformer | TiDE  | PatchTST |
> | ---------------- | --------- | ----------- | ------------ | ----- | -------- |
> | Traffic_96       | 0.435     | 0.522       | **0.395**    | 0.805 | 0.462    |
> | Traffic_192      | 0.446     | 0.530       | **0.417**    | 0.756 | 0.466    |
> | Traffic_336      | 0.461     | 0.558       | **0.433**    | 0.762 | 0.482    |
> | Traffic_720      | 0.494     | 0.589       | **0.467**    | 0.719 | 0.514    |
> | Traffic_AVG      | 0.459     | 0.550       | **0.428**    | 0.760 | 0.481    |
> | Solar-Energy_96  | **0.200** | 0.310       | 0.203        | 0.312 | 0.234    |
> | Solar-Energy_192 | **0.231** | 0.734       | 0.233        | 0.339 | 0.267    |
> | Solar-Energy_336 | 0.250     | 0.750       | 0.248        | 0.368 | 0.290    |
> | Solar-Energy_720 | **0.249** | 0.765       | **0.249**    | 0.370 | 0.289    |
> | Solar-Energy_AVG | **0.232** | 0.639       | 0.233        | 0.347 | 0.270    |
>
> For the Solar-Energy dataset, our model surpasses the current best baseline on the MSE metric.  For the Traffic dataset, which is characterized by a high number of variables, our model's performance ranks second among all baselines, just behind the iTransformer specifically designed for multivariable time series. In the future, we will further consider modeling the relationships between variables.
>
> # W4:Missing code
>
> Thank you for your reminder. In this supplementary material, we have submitted our model's code, as well as the training code. The complete repository of our method will be made open-source in the future.

---

> ### Author Response · Authors · 2024-11-25
> **Response to Reviewer pp94**
>
> Dear Reviewer pp94,
>
> We would like to express our sincere gratitude for your time and efforts in reviewing our paper.
>
> We have made an extensive effort to try to address your concerns. In our response:
>
> - We provide a detailed explanation of the specific definition of style.
>
> - We elucidate the rationale behind using images for time series forecasting.
>
> - We supplement our findings with results from our model on the Traffic and Solar Energy datasets, along with comparisons to other baseline models.
>
> - We also provide the code used in our research.
>
> We hope our response can address your concerns. If you have any further concerns or questions, please do not hesitate to inform us, and we will be more than happy to address them promptly.
>
>
> All the best,
>
> Authors

---

### Note · Authors · 2024-12-05

**Comment:**

Dear Reviewers,

Thank you for your valuable feedback and insights regarding our manuscript. We truly appreciate the time and effort you invested in reviewing our work. Your comments have provided us with a clearer direction for improvement.

We have decided to withdraw the paper for now, as we plan to address the concerns raised and enhance the quality of our research. We are committed to refining our work and hope to resubmit in the future.

Thank you once again for your constructive criticism.

Best regards, The Authors

**Withdrawal Confirmation:**

I have read and agree with the venue's withdrawal policy on behalf of myself and my co-authors.